# Quantitative RNA pseudouridine landscape reveals dynamic modification patterns and evolutionary conservation across bacterial species

Letong Xu[1†], Shenghai Shen[2†], Yizhou Zhang[1], Zhihao Guo[3], Beifang Lu[1], Jiadai Huang[1], Runsheng Li[4,5], Yitong Shen[2], Li-Sheng Zhang[2,6]*, Xin Deng[1,3,4]*

[1]Department of Biomedical Sciences, City University of Hong Kong, Hong Kong, China; [2]Division of Life Science, The Hong Kong University of Science and Technology, Hong Kong, China; [3]Shenzhen Research Institute, City University of Hong Kong, Shenzhen, China; [4]Tung Biomedical Sciences Center, City University of Hong Kong, Hong Kong, China; [5]Department of Infectious Diseases and Public Health, City University of Hong Kong, Hong Kong, China; [6]Department of Chemistry, The Hong Kong University of Science and Technology, Hong Kong, China

*For correspondence:
zhangls@ust.hk (L-SZ);
xindeng@cityu.edu.hk (XD)

†These authors contributed equally to this work

## eLife Assessment

This study illustrates a **valuable** application of BID-seq to bacterial RNA, allowing transcriptome-wide mapping of pseudouridine modifications across various bacterial species. The evidence presented includes **solid** data and analyses that would benefit from additional experimental validation. The work will interest a specialized audience involved in RNA biology.

**Abstract** Pseudouridine ($\Psi$) modifications are the most abundant RNA modifications; however, their distribution and functional significance in bacteria remain largely unexplored compared to eukaryotic systems. In this study, we present the first transcriptome-wide and quantitative mapping of $\Psi$ modifications across five diverse bacterial species (*Bacillus cereus*, *Escherichia coli*, *Klebsiella pneumoniae*, *Pseudomonas aeruginosa*, and *Pseudomonas syringae*) at single-base resolution, utilizing the optimized baBID-seq method for bacterial RNA. Our analysis revealed growth phase-dependent dynamics of pseudouridylation in bacterial tRNA and mRNA, particularly in genes enriched in core metabolic pathways. Comparative analysis demonstrated evolutionarily conserved features of $\Psi$ modifications, such as dominant motif contexts, $\Psi$ clustering within operons, etc. Functional analysis indicated $\Psi$ modifications affect bacterial mRNA stability, translation, and interactions with specific RNA-binding proteins in response to changing cellular demands during growth phase transitions. The integrated computational analysis on local RNA architecture was conducted to elucidate the structure-dependent $\Psi$ modifications in bacterial RNA. Furthermore, we developed an integrated deep learning framework, combining LSTM-transformer-GNN-based neural networks (pseU_NN) to capture both RNA sequence and local structure features for effective prediction of $\Psi$-modified sites. Overall, our study provides valuable insights into the landscapes of bacterial RNA $\Psi$ modifications and establishes a foundation for future mechanistic investigations on bacterial $\Psi$ functions.

## Introduction

RNA modifications are a crucial layer of post-transcriptional regulation in biological systems, with approximately 170 distinct chemical modifications identified to date (*Roundtree et al., 2017*). Pseudouridine (Ψ), often referred to as the 'fifth nucleoside', is one of the most prevalent and evolutionarily conserved RNA modifications (*Cerneckis et al., 2022*; *Rodell et al., 2024*). This abundant modification arises from a specific isomerization process in which uridine undergoes a site-specific intramolecular rearrangement (*Yu and Allen, 1959*). Ψ can thermodynamically stabilize RNA structures by enhancing base stacking and increasing the rigidity of the sugar-phosphate backbone, which helps maintain the structural folding of functional RNAs such as tRNA and rRNA (*Pan et al., 2003*; *Roovers et al., 2006*). For example, Ψ at the 55th position of tRNA, a universally conserved modification in both eukaryotes and prokaryotes, is crucial for regulating tRNA stability and aminoacylation (*Ishida et al., 2011*; *Schultz et al., 2024*). Ψ modifications in rRNA play important roles in rRNA biogenesis and function, as well as in mRNA translation, for both mammals and bacteria (*Leppik et al., 2017*; *Sloan et al., 2017*; *Zhao et al., 2023*).

The regulation of mRNA translation could be highly complicated, and mRNA stability remarkably impacts gene expression in mammals. Ψ-modified mRNAs demonstrate enhanced stability due to their resistance to RNase L-mediated degradation (*Anderson et al., 2011*). Pre-mRNA is found to be pseudouridylated co-transcriptionally, with Ψ enriched near alternative splicing regions and RNA-binding protein (RBP) binding sites (*Martinez et al., 2022*). Moreover, Ψ located within exon regions can alter codon properties to modulate translation, while Ψ modifications at stop codons promote ribosomal readthrough (*Hoernes et al., 2016*; *Karijolich and Yu, 2011*). Ψ also facilitates the low-level synthesis of peptide products from individual mRNA sequences in human cells and increases the rate at which near-cognate tRNA[Val] interacts with ΨUU codons (*Eyler et al., 2019*), suggesting a more complex regulatory role for Ψ in translation.

Until recently, comprehensive investigations of bacterial RNA pseudouridylation have been limited due to technical challenges in precisely mapping and quantifying Ψ modifications at single-nucleotide resolution. While eukaryotic mRNA can be easily isolated and assessed using polyA+ enrichment methods, bacterial transcripts lack polyA tails and are predominantly composed of ribosomal RNA, which accounts for over 95% of total RNA (*Liang et al., 2000*). This methodological gap has hindered the functional exploration of Ψ roles in bacterial RNA. A previously reported CMC-based method can selectively label Ψ sites and induce truncation signatures during reverse transcription (RT). However, this technical strategy could exhibit several drawbacks, including relatively low sensitivity and limitations in quantifying Ψ modification levels. Recently, a new technique named Bisulfite-Induced Deletion sequencing (BID-seq) has emerged, utilizing unique deletion signatures induced at Ψ-modified sites to achieve base-resolution and quantitative characterization of Ψ sites transcriptome-wide (*Dai et al., 2023*; *Zhang et al., 2024*).

To address these challenges and enable future functional study of bacterial Ψ modifications, we developed an optimized BID-seq method for bacterial RNA, termed baBID-seq. We selected *Klebsiella pneumoniae*, *Bacillus cereus*, *Pseudomonas aeruginosa*, and *Pseudomonas syringae* based on their biological relevance and taxonomic diversity. *K. pneumoniae*, *B. cereus*, and *P. aeruginosa* are clinically important human pathogens responsible for many infectious diseases, yet transcriptome-wide pseudouridylation has not been systematically characterized in these organisms (*Ehling-Schulz et al., 2019*; *Kerr and Snelling, 2009*; *Wyres et al., 2020*). *P. syringae*, a well-studied plant pathogen, was included to extend the analysis beyond human pathogens and to explore pseudouridine modification in a distinct ecological context (*Xin et al., 2018*). Collectively, these species encompass both Gram-positive (*B. cereus*) and Gram-negative (*K. pneumoniae*, *P. aeruginosa*, and *P. syringae*) bacteria and exhibit substantial differences in genome size, GC content, and pathogenic lifestyle. This selection provides a comparative framework for investigating conserved and species-specific features of bacterial pseudouridylation across diverse lineages. By combining efficient rRNA depletion in baBID-seq, we expanded our quantitative Ψ analysis to these representative bacterial species, examining both exponential and stationary growth phases. Our analysis revealed the landscape of Ψ modifications on bacterial rRNA, tRNA, and mRNA, highlighting evolutionarily conserved Ψ features across bacterial strains. We investigated the sequence and structural properties of local mRNA regions that influence Ψ deposition. Dynamic Ψ modifications at specific sites in tRNA and mRNA were observed, showing distinct accumulation patterns during the stationary growth phase. In *P. syringae*, we explored the roles

of $\Psi$ under nutrient-deficient conditions and found a positive correlation between mRNA translation efficiency (TE) and $\Psi$ intensity (*Hua et al., 2024*). In *P. aeruginosa*, our data suggested the potential regulatory functions of $\Psi$ in promoting mRNA interactions with the Hfq chaperone (*Trouillon et al., 2022*). Furthermore, we employed a hybrid LSTM-attention-based graph neural network (GNN) classification approach, integrating RNA sequence and local structure features to predict $\Psi$ modification sites. Collectively, our analysis revealed a dynamic landscape of $\Psi$ modifications, uncovering their evolutionarily conserved features, alongside key motif contexts and structural elements that impact $\Psi$ installation on bacterial RNA.

## Results

### baBID-seq quantitatively maps $\Psi$ modification in bacterial rRNA, tRNA, and mRNA

To investigate $\Psi$ modifications in bacteria, we primarily applied the standard BID-seq protocol (*Zhang et al., 2024*) to total RNA isolated from *E. coli* and *P. aeruginosa* during exponential and stationary growth phases. $\Psi$ sites on rRNA were identified through deletion signatures at single-base resolution, and the observed deletion ratios were utilized to assess $\Psi$ modification stoichiometry. By characterizing $\Psi$ sites with significantly higher deletion ratios in 'BID-seq treated' samples compared to 'Input' samples, we detected 9 out of 10 known $\Psi$ sites and 9 conserved $\Psi$ sites, in 23S and 16S rRNA from *E. coli* and *P. aeruginosa*, respectively (*Figure 1a, b*). For example, $\Psi$ 781 in 16S rRNA of *P. aeruginosa* exhibited a distinct deletion signature in 'BID-seq treated' samples versus the input (*Figure 1—figure supplement 1c*). Although $\Psi$ is known to regulate rRNA local structure and biogenesis (*Leppik et al., 2017*), bacterial RNA BID-seq (baBID-seq) pointed out that the number of $\Psi$-modified sites is notably lower than the typical >100 $\Psi$ sites found in mammalian rRNA (*Dai et al., 2023*). We then compared the rRNA $\Psi$ fraction in *E. coli* and *P. aeruginosa* across the two growth stages, and almost all $\Psi$ sites are highly conserved, with only slight variations in deletion ratios at $\Psi$ sites (*Figure 1a, b*).

To study $\Psi$ modifications on bacterial RNA species beyond rRNA, we incorporated probe-based rRNA depletion (*Choe et al., 2021*) into our optimized protocol for baBID-seq. We carefully established fragmentation conditions to generate RNA fragments of ~60–70 nt in length, ensuring adaptor ligation efficiency; meanwhile, size selection of the amplified library by PAGE gel minimized contamination from adaptor dimers or DNA of unexpected sizes (*Figure 1—figure supplement 1a*). We then applied the baBID-seq protocol to four bacterial species (*K. pneumoniae*, *B. cereus*, *P. aeruginosa*, and *P. syringae*). Due to rRNA depletion, rRNA-derived reads may not be suitable for comprehensive rRNA $\Psi$ profiling. However, for benchmarking baBID-seq library quality, certain 16S rRNA $\Psi$ sites with stable modification fractions can be used as internal indicators of baBID-seq performance (*Figure 1—figure supplement 1b*).

baBID-seq further successfully captured various RNA species, including rRNA, tRNA, and mRNA. We characterized hundreds of $\Psi$ sites on rRNA-depleted RNA isolated from four bacterial strains across exponential and stationary growth phases. The results from baBID-seq quantitatively demonstrated a strong correlation of deletion ratios at all detected $\Psi$ sites, between biological replicates (*Figure 1c*, *Figure 1—figure supplement 2a*). Among these, while $\Psi$ sites on rRNA and tRNA consistently showed stable modification levels across biological replicates, mRNA $\Psi$ sites displayed greater variability.

Given the conservation of tRNA $\Psi$ sites across biological replicates, we used the average $\Psi$ fraction at each specific tRNA $\Psi$ site for downstream analysis. baBID-seq quantitatively maps $\Psi$ modifications at various positions within tRNA, including the stem and loop of the T-arm, anticodon arm, and D-arm (*Figure 1d* and *Figure 1—figure supplement 2b, c*). To investigate tRNA $\Psi$ dynamics during exponential versus stationary growth phases, we quantified $\Psi$ fraction differences at each specific site across four strains. Most $\Psi$ sites on bacterial tRNA consistently exhibited higher modification fractions in stationary phase, across all examined strains (*Figure 1e*). In *K. pneumoniae*, the $\Psi$ sites within the T-arm, D-arm, and anticodon arm concordantly showed a reduced modification fraction in exponential phase, compared to stationary phase (*Figure 1d*). Similarly, in *B. cereus* and *P. syringae*, most tRNA $\Psi$ sites within the T-arm displayed lower $\Psi$ fractions under exponential phase (*Figure 1—figure supplement 2b, c*). Previous research has shown that the T-arm globally influences tRNA maturation and regulates translation in *E. coli* (*Schultz et al., 2024*). Thus, this growth phase-dependent pattern

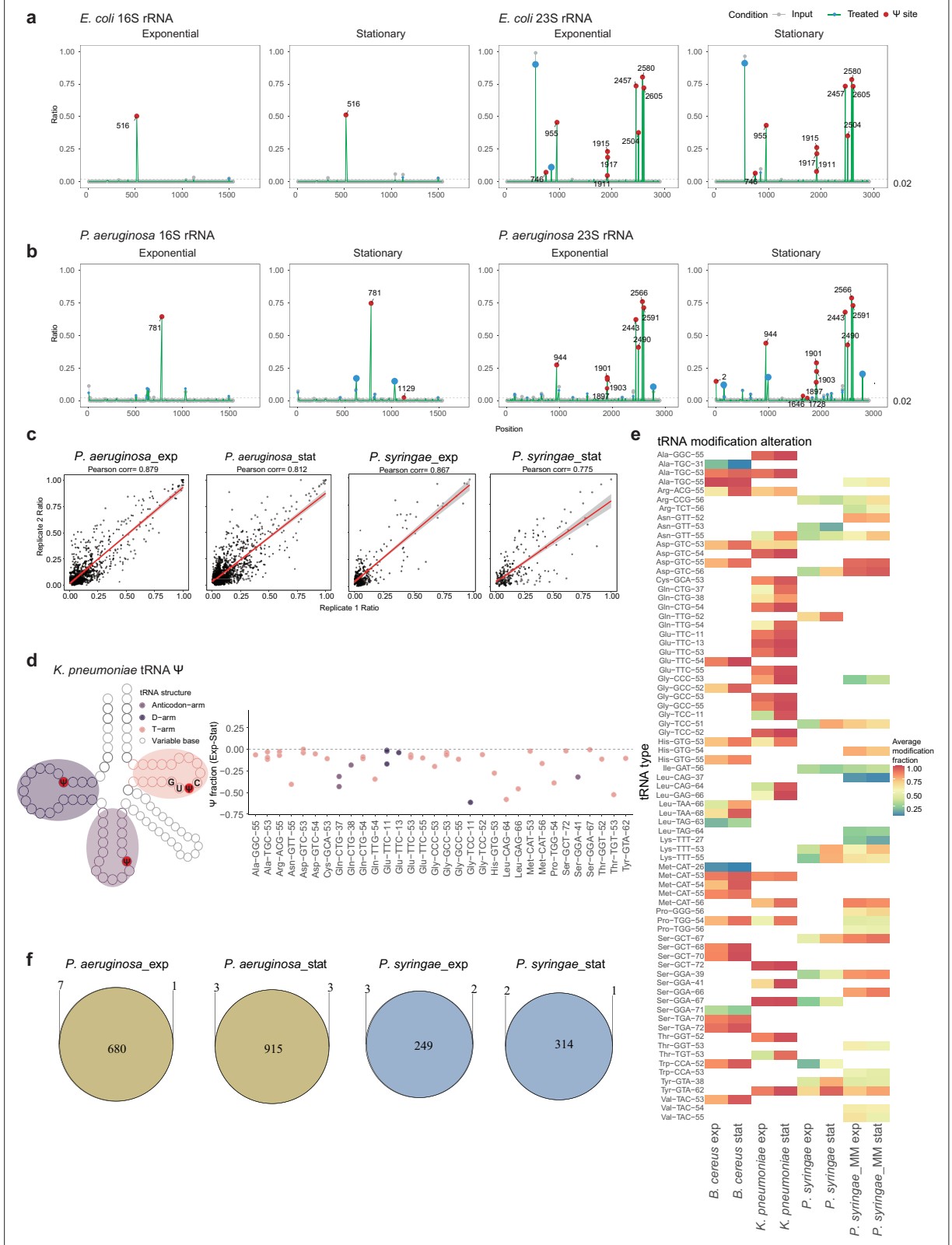

**Figure 1.** BID-seq identifies precise pseudouridine (Ψ) modification sites in ribosomal RNA and reveals dynamic Ψ modification patterns in transfer RNA. (**a, b**) Ψ modifications detected on 16S and 23S ribosomal RNA with baBID-seq of *E. coli* (**a**) and *P. aeruginosa* (**b**) total RNA during exponential and stationary growth phases. All Ψ sites in panels (**a**) and (**b**) were identified using filtration criteria of deletion fraction >0.02 and p-value ≤1 × 10⁻⁴ (**c**) Pearson correlation analysis of Ψ modification fractions at individual sites between biological replicates. (**d**) Ψ modification fraction alteration pattern

*Figure 1 continued on next page*

*Figure 1 continued*
observed on specific sites in *K. pneumoniae* different tRNA regions. The left panel depicts the general tRNA secondary structure in *K. pneumoniae*. Structural regions are indicated in the legend and highlighted with corresponding colors. The right scatter plot's *y*-axis depicts the Ψ fraction difference, calculated as (exponential phase Ψ fraction) – (stationary phase Ψ fraction). (**e**) Heatmap displaying the tRNA Ψ fraction alteration features detected across different conditions. Color represents the average Ψ fraction values (ranging from 2% to 100%) at specific sites within tRNA isoacceptors for each strain. The tRNA tags (labeled in *y*-axis) comprise the amino acids transferred by each tRNA, the corresponding anticodons, and the Ψ position on the tRNA molecules. (**f**) Venn plot shows the overlap of detected Ψ sites between biological replicates for *P. aeruginosa* and *P. syringae* during each growth phase.

The online version of this article includes the following figure supplement(s) for figure 1:

**Figure supplement 1.** baBID-seq workflow and detection of Ψ sites on 16S rRNA.

**Figure supplement 2.** Reproducibility of Ψ detection and dynamic Ψ modification in tRNA.

of tRNA pseudouridylation suggests a coordinated regulatory mechanism that may fine-tune mRNA translation as bacteria adapt to changing environmental conditions.

In addition to rRNA and tRNA, baBID-seq also identified highly conserved Ψ sites on mRNA, between biological replicates (*Figure 1f*, *Figure 1—figure supplement 2d*), providing strong evidence in identifying genuine Ψ modifications. To further verify site detection reliability, four Ψ sites were tested with pseU-TRACE (*Fang et al., 2024*): a site at position 944 on 23S rRNA, a site within the *clpV1* gene, an intergenic site located between *guaA* and *guaB* genes in *P. aeruginosa*, as well as a negative control site located within the *guaA* gene. All three positive sites were successfully detected by pseU-TRACE, while no signal was observed for the negative control (*Figure 1—figure supplement 2e*). For subsequent analysis, we focused exclusively on mRNA Ψ sites that were consistently detected across biological replicates.

## BID-seq profiles abundant Ψ modifications in bacterial mRNA

With the identification of highly conserved Ψ modifications in bacterial mRNA enabled by baBID-seq, we proceeded to analyze their distribution patterns and quantitative features transcriptome-wide. In total, we detected over 3000 Ψ sites in the mRNA of four bacterial strains. Notably, the metagene plot revealed that most Ψ sites were enriched within the coding sequences (CDS), exhibiting a remarkable consistency across strains (*Figure 2a, b*) and a similar pattern to the observations in mammals (*Dai et al., 2023*). Overall, the average Ψ modification in bacterial mRNA (mean Ψ fraction: 15%) was lower than that in 16S and 23S rRNA (mean Ψ fraction: 40%). Most mRNA Ψ sites were primarily distributed below a 25% Ψ fraction, while a smaller proportion reached fractions above 50% as highly modified sites (*Figure 2c*, *Figure 2—figure supplement 1a*).

Since Ψ modifications in mRNA untranslated regions (UTR) affect mRNA processing in eukaryotic cells (*Martinez et al., 2022*; *Rodell et al., 2024*), and UTR plays functional roles in post-translational regulation in bacteria (*Adams et al., 2021*), we conducted a detailed investigation of Ψ modifications in the upstream and downstream UTRs. UTR pseudouridylation accounted for 7.0–16.4% of total Ψ sites across the transcriptome, in both stationary and exponential phases (*Figure 2—figure supplement 1b, c*). Notably, in *P. aeruginosa* and *B. cereus*, we observed a significantly higher modification fraction for Ψ sites within downstream regions compared to CDS, in both growth phases (*Figure 2d*, *Figure 2—figure supplement 1d*). In contrast, *P. aeruginosa* exhibited a higher Ψ modification level in the upstream regions of mRNA during the exponential phase (*Figure 2d*).

We defined a Ψ-modified gene as any mRNA containing one or more Ψ sites in any growth phase and identified both phase-shared and phase-unique genes across the four strains (*Figure 2e*). We also noted that individual genes frequently harbored multiple Ψ sites, with the number of sites varying dynamically across bacterial growth phases (*Figure 2f*).

## Motif contexts surrounding Ψ modifications in bacterial mRNA, rRNA, and tRNA

Previous research has shown that different RNA species exhibit unique Ψ modification patterns, with Ψ synthases targeting specific sequence motifs in tRNA and rRNA (such as the RluA motif ΨURAA) (*Pan et al., 2003*; *Schaening-Burgos et al., 2024*). To identify the sequence determinants of Ψ modifications and uncover RNA-type-specific Ψ motif contexts across bacterial transcriptomes, we

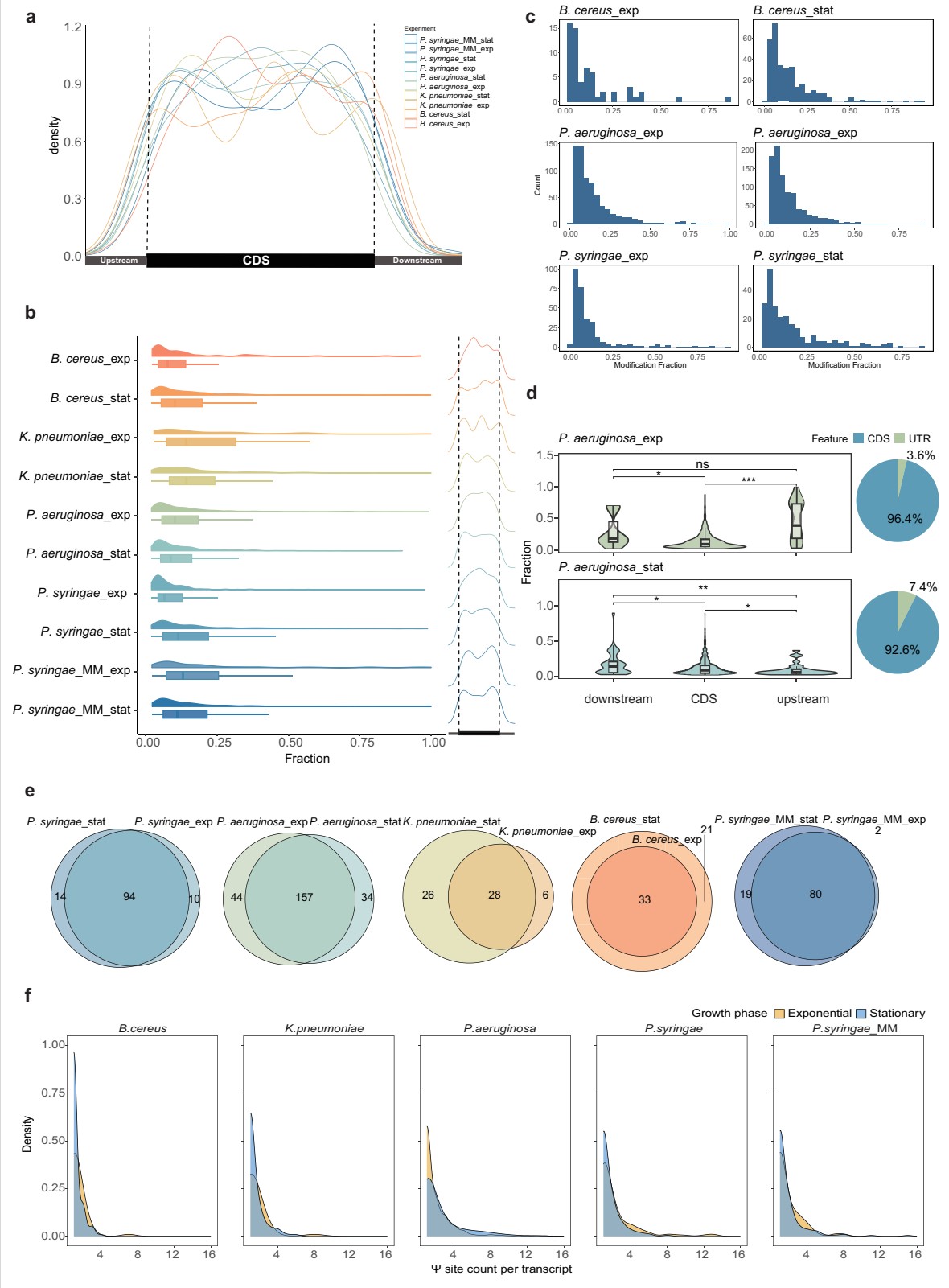

**Figure 2.** baBID-seq uncovers Ψ modification in bacterial mRNA CDS and untranslated regions (UTRs). (**a**) Density plot depicting the distribution of Ψ modifications in mRNA across different growth phases and conditions. (**b**) Distribution of mRNA Ψ fraction showing strain and growth phase-specific patterns of Ψ distribution (right Ψ density plot across each strain's mRNA). (**c**) mRNA Ψ fraction and counts under different conditions. (**d**) Right pie charts show the proportion of Ψ sites in UTRs versus coding regions, and violin plots compare Ψ fraction values between UTRs (upstream and

*Figure 2 continued on next page*

*Figure 2 continued*

downstream UTRs) and coding regions. Statistical significance was determined using the Wilcoxon signed-rank test; ns, p-value ≥0.05; *p-value <0.05; **p-value <0.01; ***p-value <0.001; and ****p-value <0.0001. (**e**) The pie charts show the Ψ-modified gene overlap in two growth phases across four strains. (**f**) Density plot shows the Ψ site numbers per transcript.

The online version of this article includes the following figure supplement(s) for figure 2:

**Figure supplement 1.** Distribution and fraction of mRNA Ψ modifications across transcript regions.

conducted a motif analysis of Ψ-modified sites with a fraction above 2%, which were confidently identified in either growth phase. The sequence context analysis focused on 5-nucleotide motifs centered at each Ψ site. We calculated the frequency of Ψ motifs through comparing the count of each unique Ψ motif versus all Ψ motifs detected in mRNA, for a single bacterial strain (*Figure 3a*).

baBID-seq analysis revealed diverse Ψ motifs within bacterial mRNA. Notably, GCΨCG, GGΨCG, and CCΨCG were the most abundant motifs observed in the three Gram-negative bacteria, while GUΨGU and GGΨGU were the dominant motifs in *B. cereus* (*Figure 3a*). For quantitative features of Ψ sites within diverse motif contexts, the average Ψ fractions for different motifs ranged from 3.4% to 96.6%, indicating varying Ψ installation efficiencies in the presence of different Ψ synthases (*Figure 3b*). Overall, we summarized the top 10 frequent Ψ motifs for bacterial mRNA of *P. aeruginosa* and *P. syringae*: GUΨCG, (CC/CU/GC/GG/UC)ΨCG, (CU/GC/UC)ΨCC, and GCΨGG (*Figure 3c*).

baBID-seq also reveals Ψ motif contexts in bacterial rRNA and tRNA. While a variety of Ψ motifs were identified in bacterial rRNA, GUΨCG motif stands out as the predominant one in tRNA across all tested strains (*Figure 3—figure supplement 1a–d*). Notably, GUΨCG motif is well-characterized within T-arm of tRNAs (at position 55) and specifically modified by TruB family (*Dai et al., 2023*; *de Crécy-Lagard et al., 2019*; *Hoang and Ferré-D'Amaré, 2001*; *Pan et al., 2003*; *Schultz et al., 2024*; *Veerareddygari et al., 2016*). According to baBID-seq data, GUΨCG has been confirmed as the predominant motif in both tRNA and rRNA (mean fraction: 68%), as well as in mRNA (mean fraction: 19%) (*Figure 3c, Figure 3—figure supplement 1c, d*), suggesting a broad role of TruB across bacterial RNA species. Several other key motifs, including UUGC, UUGA, and UUAAA, correspond to the previously characterized RluA motif in *E. coli* (*Schaening-Burgos et al., 2024*). Overall, no distinct sequence motifs were universally enriched in bacterial mRNA across strains (*Figure 3—figure supplement 1e–h*), likely due to the complex interactions among multiple Ψ synthases involved in U-to-Ψ conversion.

To determine sequence preferences of Ψ modifications under varying growth conditions, we calculated both motif frequency and average Ψ fraction for each Ψ motif in *P. aeruginosa*. Notably, we observed a decrease in GCΨCG motif frequency (*Figure 3d*), as the most abundant Ψ motif among the three Gram-negative bacteria. For other top Ψ motifs, the associated modification levels showcased slight variations between the exponential and stationary phases. To gain insights into the sequence context features around Ψ modifications, we analyzed the nucleotide composition within a 10-nt window flanking Ψ sites on bacterial mRNA. While most strains exhibited non-unique differences in GC content, *B. cereus* displayed a notably higher GC ratio (normalized to the genomic background GC content), compared to the other three strains (*Figure 3—figure supplement 1i–k*). To decipher this distinct pattern in *B. cereus*, we conducted comparative orthology-based analyses for pseudouridine synthases among five bacterial strains (*Figure 3—figure supplement 1l*). A unique interactive pattern of pseudouridine synthases in *B. cereus* may explain its divergent sequence context pattern nearby Ψ sites, compared to other strains.

## Evolutionary conservation of clustered Ψ modifications in bacterial orthologous genes

To investigate the evolutionary conservation of mRNA Ψ modifications among bacterial strains, we analyzed orthologous genes, focusing on both the preservation of modification sites and the functional characteristics of Ψ-modified genes. Based on clustered Ortho groups and genome annotations, our results revealed 225 homologous genes carrying Ψ modifications in at least two bacterial strains (*Figure 4a*). We then characterized the biological functions of these homologous Ψ-modified genes among four bacterial strains, through pathway enrichment analysis (Kyoto Encyclopedia of Genes and Genomes, KEGG), uncovering three distinct but interlinked metabolic clusters. The

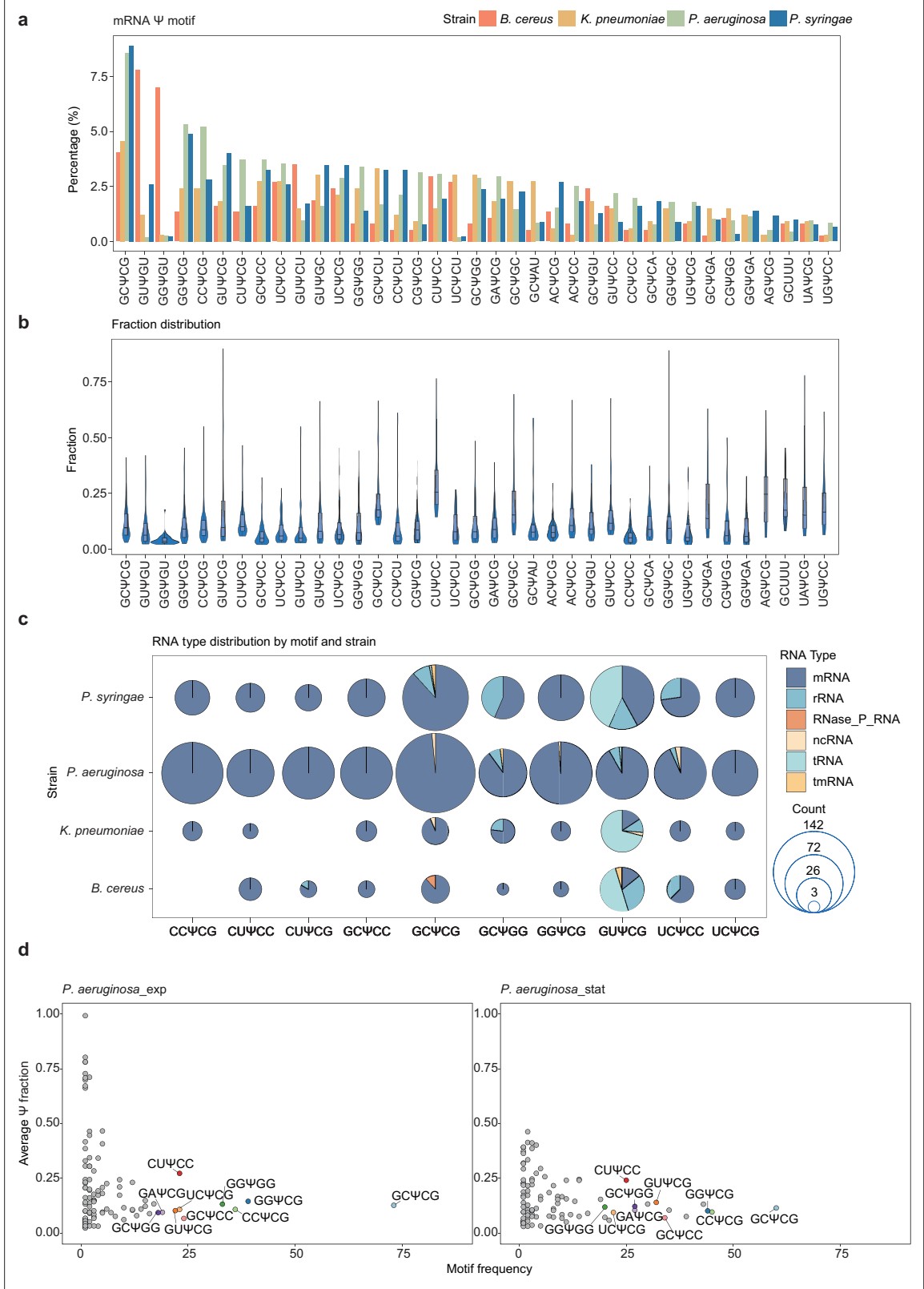

**Figure 3.** Comparative analysis of Ψ modification motif across strains. (**a**) Comparison of overall mRNA 5-mer Ψ motif ratios across four strains. Motif ratios are calculated by dividing the count of each specific 5-mer motif centered on Ψ by the total number of motifs detected in each individual strain mRNA. Ψ-modified sites with a fraction above 2% are used here. (**b**) Distribution of Ψ fraction (ranging from 2% to 100%) for each motif detected. (**c**) Scatter pie chart shows the proportional distribution of top 10 (ranked by motif abundance) Ψ-containing motif counts categorized by RNA types and

*Figure 3 continued on next page*

*Figure 3 continued*

bacterial strains. (**d**) The scatter plot illustrates the relationship between the average modification fraction and abundance of motifs in *P. aeruginosa* in exponential (*P. aeruginosa*_exp) and stationary (*P. aeruginosa*_stat) growth phases. The average Ψ fraction was calculated as the sum of Ψ fractions for each individual motif divided by its frequency.

The online version of this article includes the following figure supplement(s) for figure 3:

**Figure supplement 1.** Motif pattern of Ψ modification.

predominant cluster showed highly significant enrichment in central carbon metabolism, including glycolysis, TCA cycle, oxidative phosphorylation, and amino acid biosynthesis (*Figure 4a*, *Figure 4—figure supplement 1a*). Notably, gene clusters essential for ATP production, such as *atpA* and *atpD*, were also enriched. Among the four bacterial strains, the functional enrichment of Ψ-modified genes in core metabolic pathways, combined with the conservation of Ψ motif contexts (*Figure 3a*), provides evidence for the functional importance and evolutionary conservation of a specific group of Ψ-modified transcripts across bacterial systems. In addition to the homologous Ψ-modified genes, we identified 633 strain-specific Ψ-modified mRNAs, highlighting the dynamic nature of Ψ modifications and suggesting potential strain-specific regulatory mechanisms.

We then conducted a functional investigation of Ψ-modified genes in the two growth phases of *P. aeruginosa*. The Gene Ontology (GO) enrichment analysis revealed that Ψ-modified genes were significantly enriched in GO terms related to energy metabolism under exponential phase, compared to stationary phase, highlighting distinct state-specific patterns (*Figure 4—figure supplement 1b*). The exponential phase is known to exhibit elevated levels of metabolism-related genes, reinforcing our findings regarding the biological relevance of Ψ modifications. Notably, Ψ-modified genes showed a preferential enrichment in type IV swarming motility during stationary phase. For instance, the key transcription factor *algR*, which regulates multiple virulence factors and promotes *P. aeruginosa* twitching motility, serves as a representative Ψ-modified gene (*Kong et al., 2015*). This finding suggests that Ψ modifications may coordinate motility behaviors under stationary growth phase, potentially linking RNA modification to adaptive virulence mechanisms upon limited nutrient availability. Besides, we observed that homologous genes exhibited more stable Ψ fraction patterns, likely due to their enriched functions in fundamental metabolic processes (*Figure 4—figure supplement 1c*).

To further explore the regional distribution of Ψ modifications, we conducted a 500-nt sliding window analysis across all detected mRNA transcripts in four bacterial strains, for identifying specific regions with multiple Ψ sites (*Figure 4—figure supplement 1d*). Further analysis revealed that most Ψ-enriched regions corresponded to operons or gene clusters. baBID-seq identified Ψ clusters containing two or more Ψ sites within operons such as the evolutionarily conserved *atp* operon (*Ventura et al., 2004*) and *eno* operon, across all tested strains (*Commichau et al., 2009*; *Salgado et al., 2000*; *Figure 4b*). This phenomenon was also observed in other homologous gene clusters or operons, such as *rpoA*, *fusA*, *groE*, and *rpc* operons (*Salgado et al., 2006*; *Figure 4—figure supplement 1e*), suggesting Ψ role in bacterial post-transcriptional regulation (*Rodell et al., 2024*). We observed distinct regional patterns of Ψ modifications, in operons and gene clusters: some Ψ sites accumulated within 3′ UTR regions of *atp* operon, for instance, in *atpD* of *P. aeruginosa* (*Figure 4b*); Ψ sites in *atpD* gene were predominantly located near stop codon in *P. aeruginosa*, whereas most Ψ gathered at the translation initiation region in *B. cereus* (*Figure 4b*). This regional analysis of Ψ clusters revealed strain-specific distribution patterns in the conserved operons. The flexible regional Ψ modifications in operons, such as *atp*, may regulate gene expression to align with bacteria-specific metabolic needs.

## Growth state-dependent dynamic Ψ modifications in bacterial mRNA

Ψ modification levels on mRNA have been reported to fluctuate under stress conditions in human cells (*Li et al., 2015*). baBID-seq also observed alterations in Ψ modifications across four bacterial strains under different growth conditions. According to a detailed comparative analysis of Ψ sites between two growth phases, we witnessed both newly emerged and diminished Ψ modification events, as well as alteration in modification fractions at conserved sites. The quantitative baBID-seq approach allowed us to pinpoint dynamic Ψ modifications in response to bacterial metabolic shifts and changes in growth states. We initially compared the distribution of modification fractions for all mRNA Ψ sites

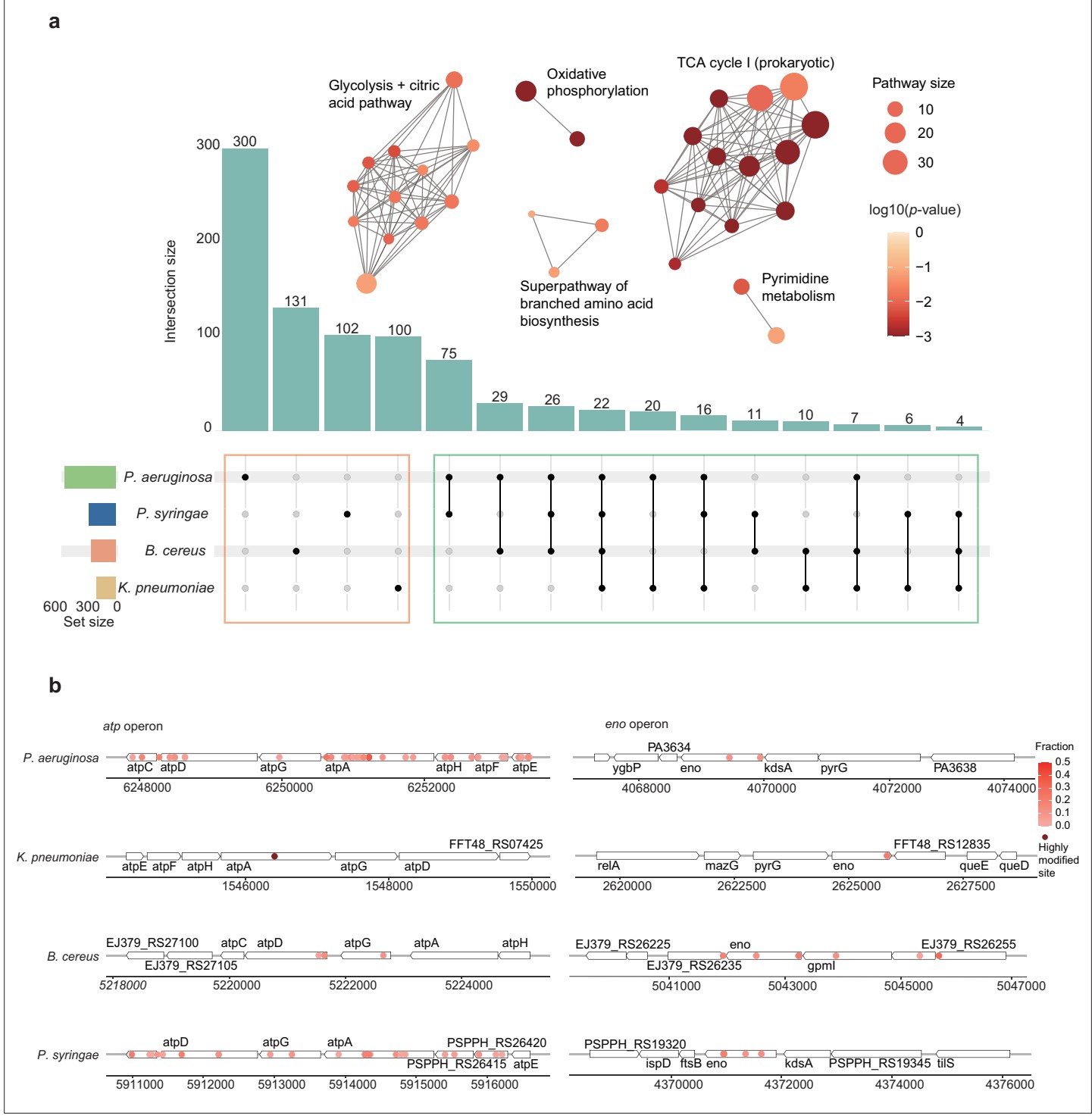

**Figure 4.** Evolutionary conservation of clustered Ψ modifications in orthologous genes. (**a**) The bar plot depicts the Ψ-modified homologous genes among bacterial species (green box) and strain-specific genes (orange box). Functional networks generated by clustering Kyoto Encyclopedia of Genes and Genomes (KEGG) pathway enrichment results of Ψ-modified homologous genes present across two or more strains. The dot size in the network indicates the gene number contained in specific KEGG pathways. The p-value for each pathway was calculated with Fisher's exact test. (**b**) Regions with Ψ enrichment across the *atp* and *eno* operons in four strains. Dark red dots represent highly modified sites with Ψ fraction values exceeding 50%.

The online version of this article includes the following figure supplement(s) for figure 4:

**Figure supplement 1.** Comprehensive analysis showing functional enrichment of Ψ-containing transcripts and Ψ-enriched region.

in exponential versus stationary phase. *K. pneumoniae* and *B. cereus* exhibited significantly higher global Ψ levels during the stationary phase (*Figure 5—figure supplement 1a, b*). In contrast, in either nutrient-enriched or minimal media (MM) condition, *P. aeruginosa* and *P. syringae* did not show significant changes in global Ψ fractions between two phases (*Figure 5—figure supplement 1c, d*).

We then focused on changes in Ψ fractions at specific sites, setting a cutoff of 10% variation between two growth phases, across four bacterial strains. In *P. aeruginosa* and *P. syringae*, we identified numerous phase-specific Ψ sites that either emerged or disappeared, as well as many conserved Ψ sites exhibiting changes in Ψ fractions above 10% (*Figure 5a, b*). For instance, in *P. aeruginosa*, key genes linked to energy metabolism, such as *sucB*, *sucC*, and *gltA*, displayed Ψ sites of increased modification fraction during exponential phase of heightened metabolic activity. The stop codon region of *secY*, a protein essential for type II secretion systems in *P. aeruginosa*, contained a highly modified Ψ site that was only detected in exponential growth phase; similar Ψ fraction dynamics were observed for *secY* in *K. pneumoniae* and *B. cereus* (*Figure 5—figure supplement 1a–c*).

In addition to analyzing Ψ sites, we also calculated Ψ intensity for each gene, defined as the sum of modification fractions for all Ψ sites across one single mRNA. During the transition from exponential to stationary phase, Ψ intensity of specific genes shifted significantly (*Figure 5c–e*, *Figure 5—figure supplement 1e*). In the exponential phase, many genes involved in metabolism, amino acid biosynthesis, and protein synthesis exhibited higher Ψ intensity, such as *rpsD*, *rpsT*, *tufB*, *lon*, and *nuoD*, reflecting their roles in supporting rapid growth; meanwhile, for these genes, their decreased Ψ intensity observed in the stationary phase may suggest a coordinated Ψ reprogramming that helps bacteria adapt to reduced nutrient availability and increased cell density, through downregulating energy-intensive processes. Overall, the dynamic nature of Ψ modifications—including newly emerged and diminished Ψ sites, as well as the ones with altered Ψ fractions between growth conditions—suggests their potential roles as responsive epitranscriptomic switches that facilitate bacteria to be adapted to varying environmental conditions and metabolic demands.

## Ψ correlates with bacterial mRNA metabolism and function

Ψ can enhance mRNA stability and translation in mammals (*Karikó et al., 2008*), parasites (*Li et al., 2025*; *Nakamoto et al., 2017*), and plants (*Li et al., 2025*). However, the effects of Ψ on bacterial mRNA have yet to be investigated. We normalized gene expression levels by calculating transcripts per kilobase million (TPM) for two growth phases and grouped the adequately expressed genes into Ψ-modified mRNA (Ψ-mRNA) and unmodified mRNA (non-Ψ-mRNA). The analysis revealed that the expression level of Ψ-mRNA was significantly higher than that of non-Ψ-mRNA in *P. aeruginosa* and *P. syringae* when cultured in a nutrient-sufficient medium (*Figure 5f, g*, *Figure 5—figure supplement 1f*). However, *P. syringae* exhibited more moderate changes in mRNA expression between two growth phases under MM conditions (*Figure 5h*). Our findings suggest that Ψ may stabilize mRNA in a growth phase-dependent manner in bacteria.

We then conducted an analysis of TE in *P. syringae* with alterations in Ψ modifications under two distinct conditions (King's B medium, KB, and MM) to examine whether Ψ impacts bacterial mRNA translation. Although we did not observe a strong global correlation between changes in Ψ modifications and TE for all genes under both conditions, a larger proportion of genes tend to show a positive correlation (*Figure 5i*). Our findings partially align with previous studies in mammals, which suggested that Ψ modifications may help enhance mRNA TE in bacteria, with complex roles to be further studied in translation regulation.

Hfq is a major bacterial post-transcriptional regulator that functions as a pivotal RBP, orchestrating various cellular processes (*Trouillon et al., 2022*). Its regulatory mechanisms have been extensively characterized, including the alteration of RNA structure and the facilitation of sRNA–mRNA interactions (*Chihara et al., 2019*), highlighting its fundamental role in coordinating gene expression networks (*Dos Santos et al., 2019*; *Sobrero and Valverde, 2012*). To investigate the potential association between Hfq and Ψ modifications, we performed an integrative analysis combining our data with previously published Hfq RIP-seq data (*Trouillon et al., 2022*) in *P. aeruginosa*. We examined both exponential and stationary growth phases to evaluate whether Hfq targets Ψ-modified regions or whether Ψ modifications affect Hfq–RNA interactions. Our results indicated that the distance between Ψ sites and Hfq peak centers significantly decreased during the stationary phase (*Figure 5—figure supplement 1g*); meanwhile, Hfq-bound genes accounted for a substantial proportion of mRNA Ψ

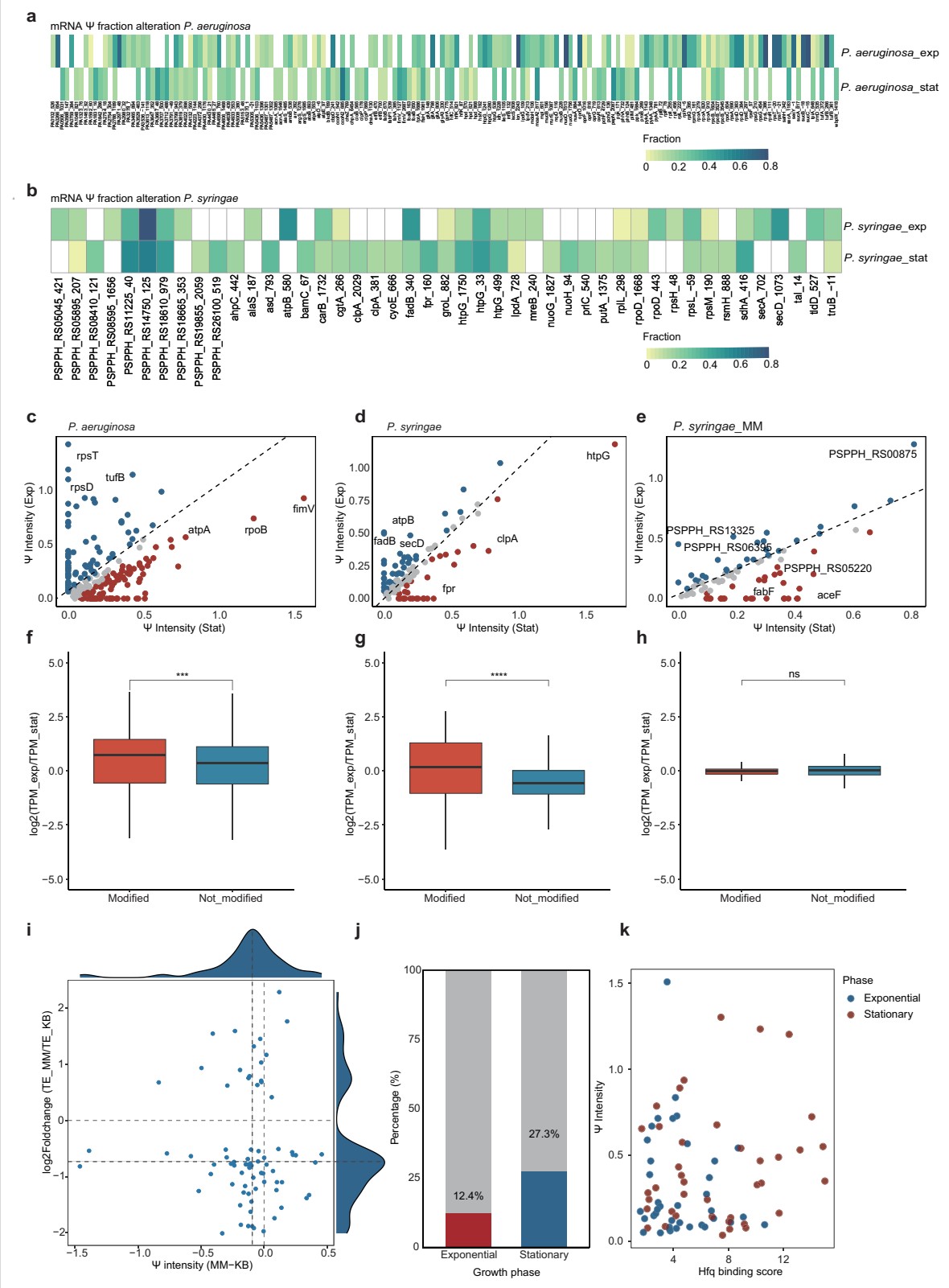

**Figure 5.** Growth state-dependent dynamics of Ψ modification fraction. Heatmap showing the mRNA Ψ fractions alteration of each site in *P. aeruginosa* (**a**) and *P. syringae* (**b**). The color intensity reflects Ψ fraction at each site. Only sites with >10% absolute difference in Ψ fraction between exponential and stationary phases are displayed. Blank boxes signify either unmodified sites or those with Ψ fractions below 2%. The annotation combining position label and gene name indicates the precise location of Ψ modification within genes. (**c**) Scatter plot shows Ψ intensity alteration

*Figure 5 continued*

between two growth phases in *P. aeruginosa*. For each mRNA, Ψ intensity is calculated as the sum of all Ψ fractions throughout the transcript. (**d**) Similar to (**c**), the scatter plot shows Ψ intensity alteration in *P. syringae* under two growth phases. (**e**) The scatter plot shows Ψ intensity alteration in *P. syringae* under two growth phases in MM medium condition. The box plot shows the modified and unmodified mRNA transcripts per kilobase million (TPM) changing between exponential and stationary growth phases under different conditions: *P. aeruginosa* cultured in Luria-Bertani (LB) medium (**f**), *P. syringae* cultured in King's B (KB) medium (**g**), and *P. syringae* cultured in MM medium (**h**). The *y*-axis shows log$_2$(TPM at exponential phase/ TPM at stationary phase) of each mRNA. The red color presents Ψ-mRNA, and the blue color indicates no-Ψ-mRNA. Wilcoxon signed-rank test; ns, p-value ≥0.05; *p-value <0.05; **p-value <0.01; ***p-value <0.001, and ****p-value <0.0001. (**i**) The scatter plot illustrates the correlation between translation efficiency (TE) alteration (log$_2$Foldchange of (TE_MM/TE_KB)) and Ψ intensity difference (Ψ intensity of MM-Ψ intensity of KB) in *P. syringae* cultured under MM medium (TE_MM) versus KB conditions (TE_KB). (**j**) The proportion of Hfq-bound Ψ-mRNA (red color for exponential growth phase mRNA and blue color for stationary phase mRNA) versus those non-Ψ-mRNA (gray color) across two growth conditions in *P. aeruginosa*. (**k**) The scatter plot shows a correlation between mRNA Ψ intensity and Hfq-binding score, where the Hfq-binding score is calculated as the sum of each mRNA peak binding strength (log$_2$Foldchange value of each peak).

The online version of this article includes the following figure supplement(s) for figure 5:

**Figure supplement 1.** Growth state-dependent dynamics of Ψ modification fraction.

sites, with 12.5% and 27.3% during exponential and stationary phases, respectively (*Figure 5j*). By defining Hfq-binding score as the sum of enrichment scores at all Hfq peaks per transcript, we found that, in stationary phase, Hfq tends to exhibit a stronger binding affinity for genes carrying more Ψ modifications (*Figure 5k*). These results suggest that Ψ modifications may facilitate mRNA–Hfq inter-actions to some extent. Overall, our results suggest that dynamic Ψ modifications could influence bacterial mRNA stability, translation, and RBP interactions, in response to altered cellular demands during growth phase transitions.

## Integrated computational analysis reveals structure-dependent Ψ modifications in bacterial RNA

We observed diverse motif contexts at Ψ sites in bacterial mRNA (*Figure 3a, b*). This observation aligns with previous studies demonstrating that pseudouridine synthases, such as PUS1 and TruB, preferentially recognize RNA local structures beyond primary sequence motifs for Ψ installation (*Carlile et al., 2019*; *Lange et al., 2012*; *Pan et al., 2003*; *Safra et al., 2017*). To computationally model the widespread Ψ modifications on bacterial mRNA, which are hypothesized to be structure-dependent, we incorporate Ψ modification determinants through clustering analyses of local RNA sequences and structural elements. We first calculated the predicted secondary structure at the 41-nt region centered by GUΨC motif, with the representative Ψ sites of 50–96% modification fraction (*Figure 6a*). Interestingly, all GUΨC motifs with varying Ψ fractions are predicted to occur within RNA loop structures. To determine whether RNA structural factors influence Ψ deposition and modification fractions, we compared two highly prevalent motifs, GUΨC and GCΨCG, by clustering predicted RNA structures across all RNA species in *P. aeruginosa*. Aside from tRNA and rRNA, which clustered together due to their distinct structural features, we observed small clusters within certain mRNAs, such as *guaB*, *recA*, and *PA4943*. Overall, no characteristic structural signatures could completely discriminate GUΨC versus GCΨCG motif (*Figure 6b*). To gain deeper insights, we conducted structural clustering analyses of all RNA species across different strains (*Figure 6—figure supplement 1a–d*). Given that some pseudouridine synthases target-specific RNA structures, we anticipated highly distinguishable clustering results; however, such distinct clustering patterns were not observed. This suggests that certain pseudouridine synthases, such as RluA (*Schaening-Burgos et al., 2024*), may not solely rely on structural features for RNA targeting. Notably, we identified clusters of Ψ sites with similar structures that exhibited higher modification fractions, including *sucC* and *sucB* in *P. aeruginosa* (*Figure 6—figure supplement 1a*), as well as *sdhA*, *PSPPH_RS14750*, and *atpB* in *P. syringae* (*Figure 6—figure supplement 1c*). Combining these findings with the distinctive Ψ fraction patterns in various motifs (*Figure 3b*), our results suggest that both RNA sequence and local structure may affect Ψ installation.

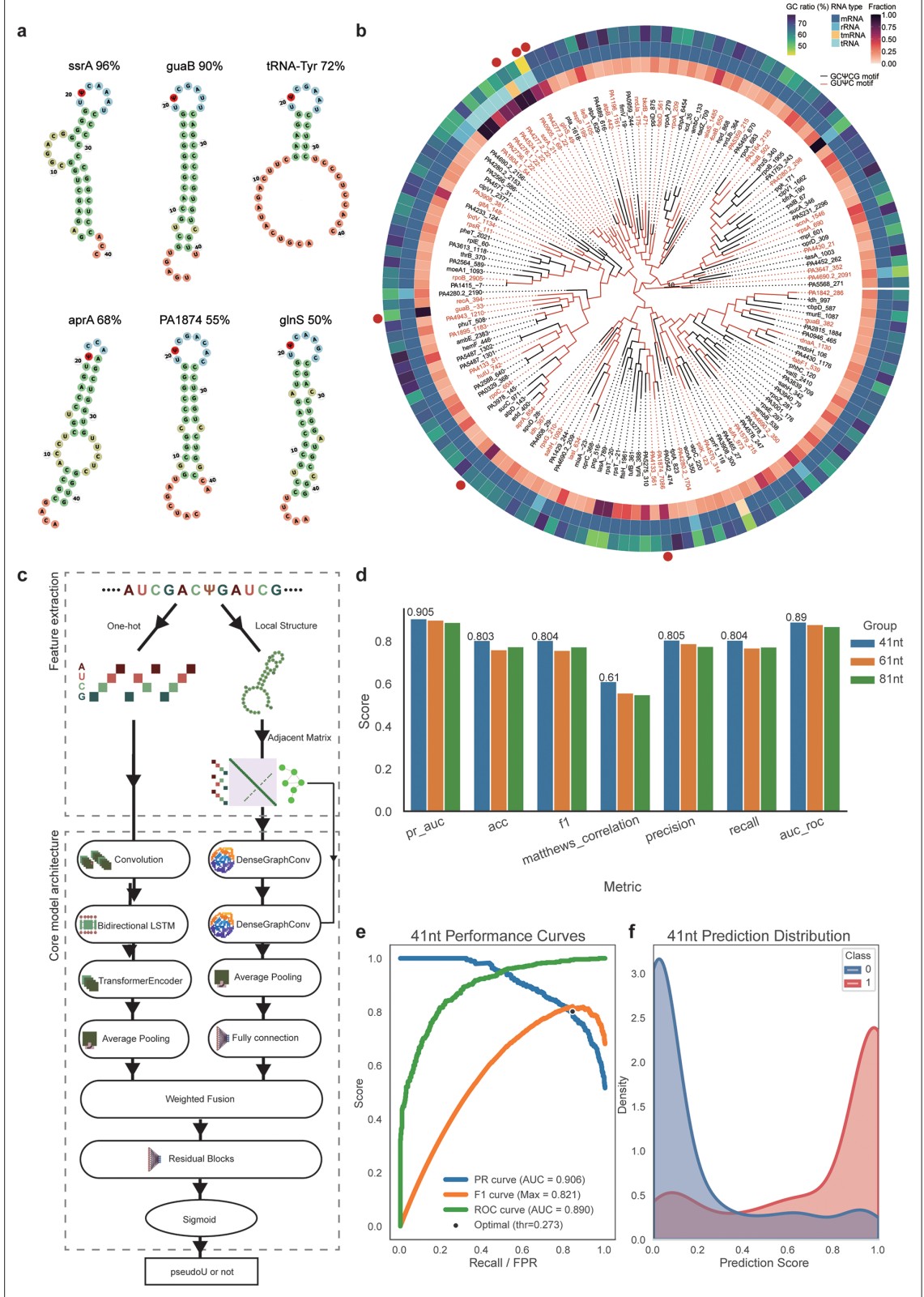

**Figure 6.** Structure-dependent patterns of Ψ modifications and transformer-graph neural network (GNN)-based deep learning network for Ψ prediction. (**a**) Predicted RNA secondary structures containing the GUΨCG motif with corresponding Ψ fraction values and gene identifiers annotated. MXfold2 is employed to model these structures using 20 nt flanking sequences extending from each modification site. (**b**) Sequence and structure clustering of 41-nucleotide RNA segments centered by Ψ sites with fraction values greater than 0.1 and containing either GCΨCG (black branch color)

*Figure 6 continued*

or GU$\Psi$C (red branch color) motifs. The circular visualization features three concentric layers: the inner layer displays the $\Psi$ fraction value, the middle layer indicates RNA type, and the outer layer represents the GC ratio (%) of each 41 nt RNA segment. The red dot around the circle marks the position of RNA displayed in (**a**). (**c**) Architecture of pseU_NN. The model integrates sequence and structural information through two parallel pathways: (1) A sequence analysis branch with one-hot embedding followed by a multi-head transformer module, and (2) A structure analysis branch that processes RNA secondary structure adjacent matrices through a graph convolution module to extract structural features. The features extracted by the two modules are further weighted and merged as input for residual blocks (fully connected layers). (**d**) Bar plots summarize model performance with input sequences of 41 nt, 61 nt, and 81 nt, evaluated by PR-AUC, accuracy (ACC), *F*1 score, Matthews correlation coefficient (MCC), precision, recall, and ROC-AUC. Overall, the three sequence lengths show comparable performance across metrics, with the 41 nt model achieving slightly higher PR-AUC and ROC-AUC, indicating that shorter sequence contexts are sufficient for robust pseudouridine prediction. (**e**) Multi-metric assessment showing precision–recall curve (AUC 0.906), *F*1 curve (AUC 0.821), and ROC curve (AUC 0.89) of the pseU_NN model on 41 nt validation datasets, achieving a peak *F*1 score of 0.804. (**f**) Distribution of pseU_NN prediction scores on 41 nt test datasets.

The online version of this article includes the following figure supplement(s) for figure 6:

**Figure supplement 1.** The structure and sequence clustering analysis of RNA molecules containing $\Psi$ modifications.

## LSTM-transformer-GNN-based neural networks for prediction of $\Psi$-modified sites

In next-generation sequencing data, variable read coverage dictated by gene expression patterns or limited sequencing depth can lead to missed $\Psi$ sites. To address this limitation, we implemented a methodology that integrates RNA sequence and local structure for a transcriptome-wide scan of $\Psi$ sites, resulting in a more comprehensive inventory of $\Psi$ candidate sites. Previous studies have shown that the sequence context surrounding $\Psi$ sites could serve as a reliable predictor (*Hoang and Ferré-D'Amaré, 2001*; *Song et al., 2021*). Building on this, we developed a deep learning model that accurately captures both sequence and structural features surrounding known $\Psi$ sites across various RNA species, allowing us to predict potential modification sites that may be condition-dependent or below baBID-seq detection thresholds. We extracted sequence segments of 41, 61, and 81 nucleotides (±20, ±30, and ±40 nt) centered at each $\Psi$ site, applying window shifts of ±5, ±10, and ±15 nt, respectively. The input sequences were then embedded using one-hot encoding and processed through a multi-head transformer module, followed by convolution layers and bidirectional LSTM (Long Short-Term Memory) layers. Simultaneously, we utilized adjacency matrices representing local RNA structures predicted using MXfold2 (*Sato et al., 2021*) as input for a GNN module. Features extracted from both modules were combined through weighted concatenation and subsequently processed using a residual block. We employed binary cross-entropy loss for predicting the likelihood of $\Psi$ modifications. This hybrid LSTM-transformer-GNN architecture effectively integrated both RNA sequence and local structure characteristics across various transcripts (*Figure 6c*), termed pseU_NN.

We used 3377 high-confidence $\Psi$ sites (with fraction values >2%) as positive samples. The negative samples consisted of 3400 randomly selected U sites that contained the unique $\Psi$ motif but without any evidence of $\Psi$ deposition in baBID-seq. The dataset was then divided into 4744 training samples, 1016 test samples, and 1017 validation samples. The model consistently performed well across different input dimensions (41, 61, and 81 nucleotides), with all variants achieving AU-ROC scores exceeding 0.8 after convergence (*Figure 6d*). Using 41-nt inputs, our approach achieved impressive validation metrics, including an area under the precision–recall curve (AU-PRC) of 0.905 and an AU-ROC of 0.89 (*Figure 6e, f*). Models trained with alternative input sequence lengths also demonstrated strong performance metrics (*Figure 6—figure supplement 1e–h*). These results set the basis for further development of effective deep learning tools for transcriptome-wide $\Psi$ prediction in bacteria and mammals.

## Discussion

RNA modifications in bacteria, particularly $\Psi$, are less characterized than their well-studied eukaryotic counterparts. Leveraging recent advances in quantitative sequencing methods such as BID-seq (*Dai et al., 2023*; *Zhang et al., 2024*), here we developed baBID-seq and presented single-base resolution maps of $\Psi$ modifications, complete with stoichiometric information, across four bacterial strains under different growth phases. Our findings confirm the widespread occurrence of $\Psi$ modifications in bacterial RNA and provide insights into their functional relevance. This extensive dataset serves as

a valuable resource for understanding the evolutionary and functional significance of Ψ modifications in bacterial RNA.

The Ψ modification plays regulatory roles in tRNA aminoacylation, stability, and the formation of functional structures (*de Crécy-Lagard et al., 2019*; *de Crécy-Lagard and Jaroch, 2021*; *Krut-yhołowa et al., 2019*; *Schultz et al., 2024*). Our analysis revealed that tRNA Ψ modifications are present in varying fractions, with a stronger modification level observed in the TΨC loop compared to the anticodon-arm and D-arm loops. Previous studies indicate that the tRNA T-arm is highly modified, not only by Ψ but also by other uridine modifications like 5-methyluridine (m⁵U) (*Chou et al., 2017*). Both Ψ and m⁵U modifications globally enhance tRNA aminoacylation and also independently influence specific tRNA modifications, such as 3-(3-amino-3-carboxypropyl)uridine at position 47 (*Schultz et al., 2024*). The TΨC loop is crucial for the interaction between tRNA and ribosome, facilitating the formation of the tRNA–ribosome complex (*Chou et al., 2017*). Interestingly, we observed a novel phenomenon where Ψ modifications on the tRNA T-arm increase in the stationary growth phase compared to the exponential phase. Given that codon composition and mRNA expression are closely correlated (*Gouy and Gautier, 1982*), dynamic Ψ modification within the TΨC loop may impact bacterial mRNA translation by modulating T-arm interactions with the ribosome. Besides, other RNA modifications on tRNA may be influenced by dynamic Ψ during growth phase transitions, potentially creating a feedback loop where existing modifications affect the biogenesis of subsequent modifications. Overall, this condition-dependent Ψ modification in tRNA may represent a new mechanism by which bacteria adapt to varying environmental conditions, anticipating future investigation.

Previous studies have demonstrated that both a deficiency and an excess of pseudouridine can severely impair ribosomal translation and proper assembly in *E. coli* (*Leppik et al., 2017*; *O'Connor et al., 2018*). The stable fraction of Ψ modification observed in *E. coli* rRNA across two different growth phases suggests that rRNA Ψ may be tightly regulated to maintain essential rRNA function. The role of Ψ in mRNA remains largely unclear across the three domains of life. Our results reveal a quantitative mRNA Ψ landscape in four bacterial strains. We found that the overall fraction of mRNA Ψ modifications was significantly lower than that of rRNA and tRNA, consistent with findings in plants and mammals (*Dai et al., 2023*; *Li et al., 2025*). The distribution and stoichiometric patterns of mRNA Ψ between Gram-positive and Gram-negative bacteria exhibited similarities. Notably, we identified evolutionarily conserved Ψ modifications in mRNAs encoding proteins involved in energy generation, ATP binding, amino acid synthesis, and protein translation, mirroring the observations in mammals and plants (*Dai et al., 2023*; *Li et al., 2025*). We also discovered clusters of multiple Ψ sites enriched in specific operons related to conserved functions, which were detected across multiple strains.

To date, limited research has focused on Ψ modifications and their alterations in bacteria. In this study, we profiled and uncovered dynamic changes in Ψ modifications during growth phase transitions among four bacterial strains. During the metabolically active phase of *P. aeruginosa*, we observed increased Ψ modifications in many metabolism-related genes. In the stationary phase, our analysis revealed reduced pseudouridylation within the CDS of *fimV*, a gene that encodes an inner membrane protein in *P. aeruginosa* responsible for regulating intracellular cyclic AMP levels, type IV-mediated twitching motility, and type II secretion system genes (*Buensuceso et al., 2016*; *Semmler et al., 2000*). Ψ modifications in CDS are known to alter codon properties on mRNA, leading to reconstituted translation and promoting the low-level synthesis of multiple peptides (*Eyler et al., 2019*). These findings suggest a potential new mechanism for regulating bacterial metabolism and quorum sensing in *P. aeruginosa*. Previous studies have demonstrated that pseudouridylation enhances mRNA stability in mammals and plants (*Dai et al., 2023*; *Li et al., 2025*; *Zhang et al., 2024*), and our research confirms and extends these observations to bacterial systems.

It has been reported that methionine aminoacyl tRNA^Met synthetase can target Ψ1074 in yeast (*Levi and Arava, 2021*), and several Ψ sites overlap with RBP-binding regions (*Martinez et al., 2022*). To systematically investigate dynamic Ψ modifications in bacterial RNA and their potential impact on RBP binding, we conducted an integrative analysis that combined our baBID-seq data with Hfq RIP-seq data from *P. aeruginosa*. Hfq is an RNA chaperone that recognizes 5-repeat AAN motifs in *P. aeruginosa* and plays a crucial role in regulating various post-transcriptional processes, including mediating sRNA–mRNA interactions (*Chihara et al., 2019*). Hfq can trigger mRNA structural reprogramming, and RNA structural switches may facilitate or suppress pseudouridylation (*Carlile et al., 2019*; *Hua et al., 2024*). We observed increased Hfq binding to Ψ-modified mRNAs during the

stationary phase compared to exponential phase, suggesting that $\Psi$ may modulate Hfq-mediated regulation via direct or indirect effects on RNA–protein affinity and mRNA structure remodeling. To establish causality, targeted perturbation of specific pseudouridine synthases or in vitro Hfq–RNA interaction assays with $\Psi$-modified versus unmodified RNAs are needed for confirmation.

$\Psi$ modification is specifically recognized in RNA local structures or motif contexts by TruB pseudouridine synthase (*Machnicka et al., 2014*). Our analysis of $\Psi$-containing sequences in mRNA revealed conserved motif signatures that closely resemble those found in tRNA and rRNA. This conservation pattern suggests that tRNA and rRNA $\Psi$ synthases may directly or indirectly recognize similar sequence and structural elements in mRNA, as evidenced by PUS1 and PUS6, which can both add $\Psi$ to tRNA and mRNA (*Carlile et al., 2019*; *Levi and Arava, 2021*). Through an integrated clustering analysis combining sequence and structural features, $\Psi$ site location and modification fraction were suggested to link with RNA local structures (*Carlile et al., 2019*). This may explain why certain potential modification sites with appropriate motifs remain unmodified or exhibit dynamic $\Psi$ levels under different growth conditions.

Deep learning methods are increasingly utilized for RNA modification prediction. Building on previous concepts, such as attention-based multi-label neural networks that predict multiple RNA modifications using sequence context (*Song et al., 2021*), we developed a hybrid LSTM-transformer-GNN architecture that integrates RNA structural features with multi-head attention mechanisms to predict potential $\Psi$ sites (pseU_NN), particularly those on transcripts of low expression levels. pseU_NN enables the prediction of potential $\Psi$ sites in different bacterial contexts, providing a more comprehensive $\Psi$ map across bacterial transcriptomes.

Overall, this study presents the first quantitative landscape of $\Psi$ modifications across diverse bacterial strains. baBID-seq revealed $\Psi$ stoichiometry in tRNA, rRNA, and mRNA under exponential and stationary growth conditions, as well as nutrient-deficient conditions. The motif analysis provides insights into pseudouridine synthase activity on bacterial mRNA. Evolutionarily conserved patterns of $\Psi$-enriched modifications were identified in operons involved in metabolic pathways across bacterial strains. In summary, our study enhances the understanding of $\Psi$ modifications and their functions in bacteria, paving the way for future mechanistic investigations.

## Materials and methods
### Bacteria strains and growth conditions
The wild-type *P. syringae pv. phaseolicola* 1448A strain was cultured in KB medium (*King et al., 1954*) (20 g/l proteose peptone, 1.5 g/l $K_2HPO_4$, 1.5 g/l $MgSO_4 \cdot 7\ H_2O$, and 10 ml/l glycerol) at 28°C for 12 hr (overnight) until reaching an optical density at 600 nm ($OD_{600}$) of 1–2, corresponding to stationary phase. *B. cereus* ATCC 14579, *P. aeruginosa* PAO1, *Escherichia coli* K-12 MG1655, and *K. pneumoniae* CR-HvKP4 were grown in Luria-Bertani (LB) broth at 37°C until achieving an $OD_{600}$ of 1–2 for stationary phase samples. For exponential phase samples, all bacterial strains were first cultured to the stationary phase as described, then subcultured into fresh medium and incubated under the same conditions until reaching an $OD_{600}$ of 0.5–0.6. For *P. syringae* in MM, exponential phase cells were harvested, washed three times with freshly prepared MM (*Huynh et al., 1989*), resuspended in MM at an $OD_{600}$ of 0.1–0.2, and cultured for an additional 6 hr.

### baBID-seq library construction
RNA was extracted using RNA isolation Kit V2 (Vazyme, #RC112-01). The extracted RNA was processed using DNase I (RNase-free, NEB # M0303S) and collected using RNA Clean & Concentrator-5 (Zymo #R1014). The RiboRID technique was used for rRNA depletion (*Choe et al., 2021*). For strains *B. cereus* and *K. pneumoniae*, rRNA was depleted with NEBNext rRNA Depletion Kit (Bacteria) (#E7850L). RNA concentration in each step was tested using Qubit RNA Assays.

One hundred nanograms of RNA generated from the ribosome removal process from each biological replicate were used for the library construction. The fragmentation was optimized to 4 min at 70°C with the fragmentation buffer used (Invitrogen). For the following steps, we strictly followed the BIDseq protocol (*Zhang et al., 2024*). The final amplified cDNA was collected and then optimal library size is selected using native PAGE gels. 40% Acrylamide/Bis (29:1) 10% native polyacrylamide gel was used. The optimal band (175–200 bp) used collected. The gel was soaked in 400 µl of 1× TE

buffer at 37°C for 1 hr on the thermal shaker at 600 rpm. The gel was crushed and snap-frozen using liquid nitrogen and incubated on a thermal shaker at 37°C at 600 rpm for 12 hr. The supernatant was collected using Spin-X, followed by DNA precipitation. The constructed libraries were sequenced on the Illumina NovaSeq sequencing platform in paired reads mode and single-end reads were used for following BID-seq data processing.

## pseU-TRACE verification

RNA samples (500 ng each for input and bisulfite-treated conditions) were subjected to bisulfite treatment using freshly prepared bisulfite (BS) reagent containing 2.4 M $Na_2SO_3$ and 0.36 M $NaHSO_3$, followed by incubation at 70°C for 3 hr. Treated RNA was purified using the Zymo RNA Clean & Concentrator-5 kit. For RT, 1 µl of 5 µM site-specific RT primer (for either the target $\Psi$ site or the negative control site) was added, and samples were incubated at 65°C for 5 min and immediately placed on ice. RT was then performed using SuperScript IV reverse transcriptase (Thermo Fisher #18090050), following the same RT conditions as in the BID-seq protocol. The resulting cDNA was treated with RNase H (NEB #M0297S) at 37°C for 20 min, followed by heat inactivation of RNase H at 70°C for 5 min. For splint-ligation, 1 µl of cDNA was mixed with upstream and downstream primers (final concentration 0.01 µM each), and the mixture was annealed using a temperature gradient (90°C for 1 min, 80°C for 1 min, 70°C for 1 min, 60°C for 1 min, 50°C for 1 min, and 40°C for 6 min). 2 µl of SplintR ligase (NEB #M0375S) was then added, and ligation was carried out at 40°C for 60 min, followed by denaturation at 95°C for 5 min and holding at 12°C. The reaction was diluted with 40 µl RNase-free $H_2O$. Quantitative real-time PCR (qPCR) was performed using a QuantStudio 3 PCR System. Each 20 µl reaction contained 2×SYBR Green qPCR Master Mix (MCE #HY-K0501), qPCR forward and reverse primers, diluted ligation product, and RNase/DNase-free water. The qPCR cycling conditions were as follows: 95°C for 5 min; 40 cycles of 95°C for 10 s and 60°C for 35 s; followed by 95°C for 15 s and 60°C for 1 min (fluorescence acquisition at a ramp rate of 0.05°C/s). Ct values were normalized to the negative control site within each replicate, and the normalized treated signal was further normalized to the corresponding input sample to quantify the $\Psi$ modification level. Primer sequences used for detection are listed in *Supplementary file 2*.

## Sequencing data processing and analysis

Sequencing data were subjected to a refined bioinformatic workflow adapted from previously established protocols. Raw sequencing reads underwent adapter trimming via Cutadapt (v.3.5) and PCR duplicate elimination using BBMap tools (v.38.73). The filtered reads were initially aligned to a curated repository of non-coding RNA sequences (including rRNAs, tRNAs, and other small RNAs). Subsequently, unmapped reads were subjected to genome alignment with optimized mapping parameters tailored for $\Psi$ detection. The resultant alignment files were processed with Samtools (v.1.13) to generate strand-specific BAM files, which were then interrogated using bam-readcount (v.1.0.1) to quantify nucleotide deletion events and calculate coverage metrics. $\Psi$ modification sites were identified by integrating deletion rate profiles, site coverage, and background signal correction derived from 'input' libraries. Bacterial samples from distinct temporal phases were analyzed as discrete entities to minimize batch effects. The quantitative assessment of $\Psi$ levels was achieved by transforming raw deletion ratios according to previously reported calibration curves, yielding high-confidence $\Psi$ fractions at single-nucleotide resolution.

To ensure robust and reliable detection of low-level $\Psi$ sites in mRNA, we applied the following stringent BID-seq filtration criteria to all candidate sites (all sites reported in the *Supplementary file 1* passed these thresholds): (1) total sequencing coverage >20 reads in both the bisulfite-treated (BID-seq) libraries ($\Sigma\, dt$ >20) and untreated input libraries ($\Sigma\, di$ >20); (2) average deletion number >5 in the treated libraries; (3) average modification fraction >0.02 (2%) in the treated libraries; and (4) average deletion ratio in treated libraries at least twofold higher than in input libraries. We consider sites with stoichiometry thresholds >0.5 as highly modified sites.

## TPM calculation

The TPM value for a specific transcript *i* is calculated as:

$$TPM_i = \left( \frac{\frac{q_i}{l_i}}{\sum_j \left( \frac{q_j}{l_j} \right)} \right) \times 10^6$$

,

$q_i$ is the number of reads mapped to transcript $i$. $l_i$ is the length of transcript $i$ (in nucleotides). $10^6$ is the scaling factor to express the result as transcripts per million, $j$ is unique transcript number (*Zhao et al., 2021*).

## Evolutionary analysis

Orthologous gene analysis was performed using OrthoFinder (*Emms and Kelly, 2019*) software to identify homologous gene clusters across the five bacterial strains. The analysis incorporated complete genome sequences and their corresponding GFF (General Feature Format) annotation files with default parameters. The resulting orthologous gene clusters were subsequently utilized for KEGG pathway analysis. Genes that were not clustered in the orthology analysis were excluded from the KEGG pathway mapping to ensure reliable results. Gene functional annotations were performed using eggNOG-mapper v2 (*Cantalapiedra et al., 2021*).

## RNA structure analysis

Pseudouridine sites with a modification fraction exceeding 2% were retained for downstream analysis. To capture the local sequence context surrounding each modification, 20 nucleotides upstream and downstream of each pseudouridine site were extracted, yielding 41-nucleotide sequences centered on the modification position. These sequences were subsequently subjected to RNA secondary structure prediction using MXfold2 (*Sato et al., 2021*) and clustering analysis using RNAclust (*Engelhardt et al., 2010*). All downstream result processing and visualization were performed using custom scripts implemented in R.

## pseU_NN

### Datasets preparation

The negative control dataset was constructed by scanning bacterial genomes for pseudouridylation-compatible sequence motifs that were absent from the experimentally verified modification sites identified in baBID-seq data. Sequence segments of 41, 61, and 81 nucleotides were then generated, with each unmodified motif positioned at the center. MXfold2 (*Sato et al., 2021*) was used to predict the secondary structure of all sequences as input for training and validation process. A total of 3377 high-confidence $\Psi$ sites with fraction values exceeding 2% were used as positive samples, while 3400 randomly selected sites carrying the unique $\Psi$ motif but showing no experimental evidence of $\Psi$ deposition were designated as negative samples. For each sequence length, the complete dataset was partitioned into 4744 training samples, 1016 test samples, and 1017 validation samples.

## Model architecture

The model architecture comprises three main components. First, two dense graph encoders were used to extract RNA secondary structure features derived from MXfold2 predictions and from one-hot-encoded RNA sequences, respectively. Second, the one-hot-encoded sequence was further processed by a convolutional layer for local sequence feature extraction, followed by two bidirectional LSTM layers to capture long-range dependencies. Positional encoding was subsequently applied before the representations were forwarded to a single-layer multi-head Transformer block. Finally, features extracted from the structure- and sequence-based modules were integrated via weighted concatenation and passed through a residual block composed of fully connected layers, with a sigmoid activation function applied to the final output to predict the probability of pseudouridine modification.

## Training and evaluation

The pseU_NN was trained using binary cross-entropy loss with the Adam optimizer. We implemented a dynamic learning rate scheduler that adjusted the rate based on validation AUC-ROC performance.

For the final evaluation, we utilized an independent test dataset, with *F*1 scores and precision–recall curves guiding prediction threshold selection. This balanced approach ensured optimal sensitivity and

specificity—critical in biological classification systems where both false positives and false negatives carry significant consequences.

The $F1$ score is the harmonic mean of precision and recall. The definition of true positive (TP), true negative (TN), false positives (FP), and false negative (FN) with shifting was adopted in previous studies (*Wang et al., 2023*; *Yu et al., 2021*).

$$precision = \frac{TP}{TP+FP},$$

$$recall = \frac{TP}{TP+FN},$$

$$F1\ score = \frac{2 \times precision \times recall}{precision + recall}.$$

The accuracy calculated using standard definitions:

$$accuracy = \frac{TP + TN}{TP + FP + TN + FN}.$$

The Matthews correlation coefficient (MCC) is calculated using the following formula:

$$MCC = \frac{TP \times TN - FP \times FN}{\sqrt{(TP + FP)(TP + FN)(TN + FP)(TN + FN)}}.$$

## Code availability

The code for pseU_NN used for this paper is available at https://github.com/Dylan-LT/pseU_NN, copy archived at *Dylan-LT, 2026*.

## Acknowledgements

This study was funded by the Guangdong Major Project of Basic and Applied Basic Research (2020B0301030005), National Natural Science Foundation of China (32172358), General Research Funds of Hong Kong (21103018, 11101619, and 11102720), and Early Career Scheme of Hong Kong (26103623). The funders had no role in study design, data collection, interpretation, or the decision to submit the work for publication.

## Additional information

### Funding

| Funder | Grant reference number | Author |
|---|---|---|
| Guangdong Major Project of Basic and Applied Basic Research | 2020B0301030005 | Xin Deng |
| General Research Funds of Hong Kong | 21103018 | Xin Deng |
| Early Career Scheme of Hong Kong | 26103623 | Xin Deng |
| National Natural Science Foundation of China | 32172358 | Xin Deng |
| General Research Funds of Hong Kong | 11101619 | Xin Deng |
| General Research Funds of Hong Kong | 11102720 | Xin Deng |

The funders had no role in study design, data collection, and interpretation, or the decision to submit the work for publication.

## Author contributions
Letong Xu, Conceptualization, Data curation, Software, Formal analysis, Validation, Visualization, Methodology, Writing – original draft; Shenghai Shen, Data curation, Software, Visualization, Methodology; Yizhou Zhang, Methodology; Zhihao Guo, Yitong Shen, Software; Beifang Lu, Data curation, Writing – original draft, Writing – review and editing; Jiadai Huang, Runsheng Li, Writing – review and editing; Li-Sheng Zhang, Writing – original draft, Writing – review and editing; Xin Deng, Supervision

## Author ORCIDs
Letong Xu https://orcid.org/0009-0001-8897-0591
Shenghai Shen https://orcid.org/0000-0001-8422-5423
Runsheng Li https://orcid.org/0000-0003-1563-1844
Xin Deng https://orcid.org/0000-0003-1580-0089

Reviewer #1 (Public review): https://doi.org/10.7554/eLife.107545.3.sa1
Reviewer #2 (Public review): https://doi.org/10.7554/eLife.107545.3.sa2
Reviewer #3 (Public review): https://doi.org/10.7554/eLife.107545.3.sa3
Author response https://doi.org/10.7554/eLife.107545.3.sa4

# Additional files

## Supplementary files
Supplementary file 1. Modification sites.

Supplementary file 2. Primer.

MDAR checklist

## Data availability
All sequencing data files have been submitted to the National Center for Biotechnology Information (NCBI) Gene Expression Omnibus (GEO) database with the reference code of GSE292335.

The following dataset was generated:

| Author(s) | Year | Dataset title | Dataset URL | Database and Identifier |
| --- | --- | --- | --- | --- |
| Deng X, Xu L | 2026 | Quantitative RNA pseudouridine maps reveal functional insights into pseudouridylation in bacteria | https://www.ncbi.nlm.nih.gov/geo/query/acc.cgi?acc=GSE292335 | NCBI Gene Expression Omnibus, GSE292335 |

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
