## [Editor Report · eLife Assessment]

This study illustrates a **valuable** application of BID-seq to bacterial RNA, allowing transcriptome-wide mapping of pseudouridine modifications across various bacterial species. The evidence presented includes **solid** data and analyses that would benefit from additional experimental validation. The work will interest a specialized audience involved in RNA biology.

---

## [Referee Report · Reviewer #1 (Public review)]

Summary:

The manuscript by Xu et al. reported base-resolution mapping of RNA pseudouridylation in five bacterial species, utilizing recently developed BID-seq. They detected pseudouridine (Ψ) in bacterial rRNA, tRNA, and mRNA, and found growth phase-dependent Ψ changes in tRNA and mRNA. They then focused on mRNA and conducted comparative analysis of Ψ profiles across different bacterial species. Finally, they developed a deep learning model to predict Ψ sites based on RNA sequence and structure.

Strengths:

This is the first comprehensive Ψ map across multiple bacterial species, and systematically reveals Ψ profiles in rRNA, tRNA, and mRNA under exponential and stationary growth conditions. It provides a valuable resource for future functional studies of Ψ in bacteria.

Weaknesses:

Ψ is highly abundant on non-coding RNA such as rRNA and rRNA, while its level on mRNA is very low. The manuscript focuses primarily on Ψ on mRNA, which is prone to false positives. Many conclusions in the manuscript are speculative, based solely on the sequencing data, but not supported by additional experiments.

---

## [Referee Report · Reviewer #2 (Public review)]

Summary:

In this study, Xu et al. present a transcriptome-wide, single-base resolution map of RNA pseudouridine modifications across evolutionarily diverse bacterial species using an adapted form of BID-Seq. By optimizing the method for bacterial RNA, the authors successfully mapped modifications in rRNA, tRNA, and, importantly, mRNA across both exponential and stationary growth phases. They uncover evolutionarily conserved Ψ motifs, dynamic Ψ regulation tied to bacterial growth state, and propose functional links between pseudouridylation and bacterial transcript stability, translation, and RNA-protein interactions. To extend these findings, they develop a deep learning model that predicts pseudouridine sites from local sequence and structural features.

Strengths:

The authors provide a valuable resource: a comprehensive Ψ atlas for bacterial systems, spanning hundreds of mRNAs and multiple species. The work addresses a gap in the field - our limited understanding of bacterial epitranscriptomics, by establishing both the method and datasets for exploring post-transcriptional modifications.

Weaknesses:

The main limitation of the study is that most functional claims (i.e. translation efficiency, mRNA stability, and RNA-binding protein interactions) are based on correlative evidence. While suggestive, these inferences would be significantly strengthened by targeted perturbation of specific Ψ synthases or direct biochemical validation of proposed RNA-protein interactions (e.g., with Hfq). Additionally, the GNN prediction model is a notable advance.

---

## [Referee Report · Reviewer #3 (Public review)]

Summary:

This study aimed to investigate pseudouridylation across various RNA species in multiple bacterial strains using an optimized BID-seq approach. It examined both conserved and divergent modification patterns, the potential functional roles of pseudouridylation, and its dynamic regulation across different growth conditions.

Strengths:

The authors optimized the BID-seq method and applied this important technique to bacterial systems, identifying multiple pseudouridylation sites across different species. They investigated the distribution of these modifications, associated sequence motifs, their dynamics across growth phases, and potential functional roles. These data are of great interest to researchers focused on understanding the significance of RNA modifications, particularly mRNA modifications, in bacteria.

---

## [Author Response]

The following is the authors’ response to the original reviews.

**Public Reviews:**

**Reviewer #1 (Public review):**
The manuscript by Xu et al. reported base-resolution mapping of RNA pseudouridylation in five bacterial species, utilizing recently developed BID-seq. They detected pseudouridine (Ψ) in bacterial rRNA, tRNA, and mRNA, and found growth phase-dependent Ψ changes in tRNA and mRNA. They then focused on mRNA and conducted a comparative analysis of Ψ profiles across different bacterial species. Finally, they developed a deep learning model to predict Ψ sites based on RNA sequence and structure.This is the first comprehensive Ψ map across multiple bacterial species, and systematically reveals Ψ profiles in rRNA, tRNA, and mRNA under exponential and stationary growth conditions. It provides a valuable resource for future functional studies of Ψ in bacteria.

We thank Reviewer 1 for the supportive and positive comments, particularly for highlighting the novelty and value of our comprehensive pseudouridine landscapes across multiple bacterial species as a valuable resource for the scientific community.

Ψ is highly abundant on non-coding RNA such as rRNA and tRNA, while its level on mRNA is very low. The manuscript focuses primarily on mRNA, which raises questions about the data quality and the rigor of the analysis. Many conclusions in the manuscript are speculative, based solely on the sequencing data but not supported by additional experiments.

We appreciate the insightful comments of Reviewer 1. We fully agree that Ψ is highly abundant on rRNA and tRNA, while its fractions on mRNA are generally lower. Ψ is highly conserved at specific positions in rRNA and tRNA, such as Ψ within tRNA T‑arm (position 55), where it plays essential roles in tRNA structural folding, tRNA stability, and mRNA translation, across plants, mammals, and bacteria[1–3]. However, most Ψ sites in mRNA exhibit lower fractions compared to rRNA and tRNA. This phenomenon is also widely observed in HeLa cell mRNA and plant mRNA, as evidenced by bisulfite-induced deletion sequencing and 2-bromoacrylamide-assisted cyclization sequencing[3–5]. In bacteria, the modifications on mRNA are harder to map and quantify, due to its low abundance in total RNA and difficulty in bacterial rRNA removal. This highlights the significance of our study.

To prove our data quality and analytical rigor, we first present the most convincing sites in bacteria, as benchmark sites. Specifically, we detected 9 out of 10 known conserved pseudouridine (Ψ) sites in *E. coli* across two biological replicates [6], displaying notable modification fraction. Ψ516 site in *E. coli* 16S rRNA, which serves as a benchmark site, consistently exhibited a high modification fraction (~100%) under multiple growth conditions, underscoring the robustness of our method. In other strains, we also observed conserved 16S rRNA Ψ sites.

To further demonstrate strong reproducibility and sensitivity. We selected three positive Ψ sites from two independent biological replicates for experimental validation, alongside one negative control site, using pseU‑TRACE method[6]. Ct values were first normalized to the corresponding Ct value of the negative control site, and the treated samples were then further normalized to their corresponding input controls (new Supplementary Fig. 2e).

Four Ψ sites were tested with pseU‑TRACE: Ψ site at position 944 on 23S rRNA, a negative control site located within *guaA* gene, a Ψ site within *clpV1* gene, and an intergenic Ψ site located between *guaA* and *guaB* genes. We successfully validated these Ψ sites in *P. aeruginosa*. The detailed pseU‑TRACE experimental procedures and corresponding data figures have been added to the revised manuscript, in either Results or Methods sections (Line 171-175, 594–617).

Previous transcriptome-wide mapping of Ψ have primarily relied on CMC-based methods to induce RT truncation signatures at the modified sites, exhibiting a limited Ψ detection sensitivity caused by low labeling efficiency[5]. In contrast, BID-seq method used in this study provides substantially higher sensitivity of Ψ detection, particularly the low-stoichiometry Ψ sites within mRNA. The high reliability and quantitative performance of BID-seq have been extensively validated in prior work using mammalian cells and synthetic Ψ-containing oligonucleotides[4].

To further ensure robustness and minimize false positives—when identifying low-level mRNA Ψ sites through bioinformatic analysis—we have applied stringent and uniform filtration criteria to all candidate sites on mRNA (new Supplementary Table 1):

(1) Total sequencing coverage >20 reads in both ‘Treated’ (BID-seq; Σd_t_ > 20) and ‘Input’ libraries (Σd_i_ > 20);

(2) An average deletion count >5 in ‘Treated’ libraries;

(3) An average modification fraction >0.02 (2%) in ‘Treated’ libraries;

(4) A deletion ratio in ‘Treated’ libraries at least two-fold higher than that in ‘Input’ libraries.

Sites with a Ψ stoichiometry >0.5 (50%) were classified as highly modified. These filtration criteria have now been explicitly described in Methods section (Lines 739–745). We strictly adhered to these Ψ site identification standards, leading to all subsequent analysis and functional studies.

Finally, to address concerns regarding reproducibility, we calculated mRNA Ψ site overlap and correlation of Ψ fractions, between two biological replicates, which has been presented in (new Supplementary Fig. 2a,d).

Overall, we have revised the manuscript to clarify these methodological strengths, and validate mRNA Ψ detection. We also tone down all speculative conclusions, with more clear linkage to the actual sequencing data, which await future functional validation.

**Reviewer #2 (Public review):**
Summary:In this study, Xu et al. present a transcriptome-wide, single-base resolution map of RNA pseudouridine modifications across evolutionarily diverse bacterial species using an adapted form of BID-Seq. By optimizing the method for bacterial RNA, the authors successfully mapped modifications in rRNA, tRNA, and, importantly, mRNA across both exponential and stationary growth phases. They uncover evolutionarily conserved Ψ motifs, dynamic Ψ regulation tied to bacterial growth state, and propose functional links between pseudouridylation and bacterial transcript stability, translation, and RNA-protein interactions. To extend these findings, they develop a deep learning model that predicts pseudouridine sites from local sequence and structural features.Strengths:The authors provide a valuable resource: a comprehensive Ψ atlas for bacterial systems, spanning hundreds of mRNAs and multiple species. The work addresses a gap in the field - our limited understanding of bacterial epitranscriptomics, by establishing both the method and datasets for exploring post-transcriptional modifications.

We thank Reviewer 2 for the supportive and positive comments. We appreciate the reviewer’s recognition of the novelty and value of our work in providing a comprehensive pseudouridine atlas across multiple bacterial species.

Weaknesses:The main limitation of the study is that most functional claims (i.e., translation efficiency, mRNA stability, and RNA-binding protein interactions) are based on correlative evidence. While suggestive, these inferences would be significantly strengthened by targeted perturbation of specific Ψ synthases or direct biochemical validation of proposed RNA-protein interactions (e.g., with Hfq).

We thank Reviewer 2 for the constructive feedback. We fully agree that our functional claims regarding translation efficiency, mRNA stability, and RNA-binding protein interactions rely primarily on correlative evidence from existing datasets rather than a direct experimental validation. We agree that the perturbation of specific pseudouridine synthases and direct biochemical validation of proposed RNA-protein interactions (for instance, Hfq) would substantially strengthen the conclusions on bacterial Ψ function. In Discussion section, we have added a discussion on this limitation of our current study (Line 517–523). Considering the scope of our current work, we anticipate such validation experiments in future research.

Additionally, the GNN prediction model is a notable advance, but methodological details are insufficient to reproduce or assess its robustness.

In response to methodological concerns regarding our pseU_GNN prediction model, we have undertaken substantial improvements to address these issues comprehensively. We have updated the complete codebase on GitHub (https://github.com/Dylan-LT/pseU_NN.git) with comprehensive documentation and a user-friendly prediction tool specifically designed for Ψ site prediction across the four bacterial species examined in this study.

We further systematically evaluated multiple neural network architectures and implemented critical architectural refinements. Specifically, we incorporated bidirectional LSTM (bid-LSTM) layers upstream of the transformer block to more effectively capture sequential dependencies and contextual information in RNA sequences. This enhanced architecture demonstrates substantially improved predictive performance, achieving an AUC-ROC of 0.89 on independent test datasets using 41-nucleotide input sequences (new Figure 6).

We have revised Figure 6 and Supplementary Fig. 7, along with their corresponding content and figure legends (Lines 428-430, 434–436, 440-447, 1065-1073), to reflect these architectural improvements and performance enhancements. We have detailed the methods part (Lines 679–708), including model architecture, validation methods and evaluation score calculation. Additionally, we have provided detailed documentation of the evaluation score calculation methodology to ensure reproducibility and transparency.

**Reviewer #3 (Public review):**
Summary:This study aimed to investigate pseudouridylation across various RNA species in multiple bacterial strains using an optimized BID-seq approach. It examined both conserved and divergent modification patterns, the potential functional roles of pseudouridylation, and its dynamic regulation across different growth conditions.Strengths:The authors optimized the BID-seq method and applied this important technique to bacterial systems, identifying multiple pseudouridylation sites across different species. They investigated the distribution of these modifications, associated sequence motifs, their dynamics across growth phases, and potential functional roles. These data are of great interest to researchers focused on understanding the significance of RNA modifications, particularly mRNA modifications, in bacteria.

We thank Reviewer 3 for the supportive and positive assessment. We are particularly grateful for the reviewer’s acknowledgment of the value of our analyses on modification distribution, sequence motifs, growth‑phase dynamics, and potential functional roles, which we hope will be of broad interest to researchers studying bacterial RNA modifications, particularly mRNA Ψ.

Weaknesses:(1) The reliability of BID-seq data is questionable due to a lack of experimental validations.

We thank Reviewer 3 for the constructive feedback. We have undertaken comprehensive revisions to address the concerns regarding manuscript structure and information organization. We have incorporated pseU‑TRACE experiments and data quality results to provide orthogonal validation of Ψ detection, strengthening the robustness of our work.

Here we copied the response in Reviewer 1 section:

“To further demonstrate strong reproducibility and sensitivity. We selected three positive Ψ sites from two independent biological replicates for experimental validation, alongside one negative control site, using pseU‑TRACE method[6]. Ct values were first normalized to the corresponding Ct value of the negative control site, and the treated samples were then further normalized to their corresponding input controls (new Supplementary Fig. 2e).

Four Ψ sites were tested with pseU‑TRACE: Ψ site at position 944 on 23S rRNA, a negative control site located within *guaA* gene, a Ψ site within *clpV1* gene, and an intergenic Ψ site located between *guaA* and *guaB* genes. We successfully validated these Ψ sites in *P. aeruginosa*. The detailed pseU‑TRACE experimental procedures and corresponding data figures have been added to the revised manuscript, in either Results or Methods sections (Line 171-175, 594–617).”

(2) The manuscript is not well-written, and the presented work shows a major lack of scientific rigor, as several key pieces of information are missing.

We thank Reviewer 3 for the suggestion. We restructured the main text to present a clearer logical flow, with key objectives (Lines 83–96, 171–175, 428–447, 517-523) explicitly stated in Introduction section and Conclusions section, with data figures directly addressing these stated aims (Supplementary Fig. 1–7).

(3) The manuscript's organization requires significant improvement, and numerous instances of missing or inconsistent information make it difficult to understand the key objectives and conclusions of the study.

We thank Reviewer 3 for the constructive feedback. All supplementary figures have been updated with detailed figure legend, methodology description, and consistent formatting. We also systematically inspected and resolved instances of missing or inconsistent information throughout the main text and supplementary materials (Supplementary Fig. 1–7; Supplementary Table 1). To enhance computational reproducibility, we have updated our GitHub repository with well-documented code and developed user-friendly prediction tools for Ψ identification across the four bacterial species examined in this study.

(4) The rationale for selecting specific bacterial species is not clearly explained, and the manuscript lacks a systematic comparison of pseudouridylation among these species.

We thank Reviewer 3 for the constructive feedback. The bacterial species analyzed in this study were selected based on both diversity and significance. *K. pneumoniae*, *B. cereus*, and *P. aeruginosa* are top model human pathogens responsible for a wide range of clinically significant infections, yet transcriptome-wide pseudouridylation has not been systematically explored in these organisms[7–9]. *P. syringae*, the most important model plant pathogen, was included to extend our analysis beyond human pathogens and to examine Ψ modification in a distinct ecological and evolutionary context, where epitranscriptomic regulation also remains poorly characterized[10]. Importantly, the selected species represent both Gram-positive (*B. cereus*) and Gram-negative (*K. pneumoniae*, *P. aeruginosa*, and *P. syringae*) bacteria, spanning substantial differences in genome size, GC content, lifestyle, and pathogenic strategies. This diversity enables a comparative framework for examining conserved and species-specific pseudouridylation patterns across bacterial lineages.

To address the reviewer’s concern, we have revised the manuscript to more clearly articulate the rationale for species selection and have added a comparative analysis highlighting similarities and differences in Ψ site distribution and modification levels among these species (Lines 83–96). We systematically compared Ψ-carrying motif for analyzing sequence context of 10 bases flanking Ψ sites in bacterial mRNA, with Supplementary Fig. 4 added.

Reference

(1) Leppik, M., Liiv, A. & Remme, J. Random pseuoduridylation in vivo reveals critical region of *Escherichia coli* 23S rRNA for ribosome assembly. Nucleic Acids Res. 45, (2017).

(2) Rajan, K. S. et al. A single pseudouridine on rRNA regulates ribosome structure and function in the mammalian parasite *Trypanosoma brucei*. Nat. Commun. 14, (2023).

(3) Li, H. et al. Quantitative RNA pseudouridine maps reveal multilayered translation control through plant rRNA, tRNA and mRNA pseudouridylation. Nat. Plants 11, 234–247 (2025).

(4) Dai, Q. et al. Quantitative sequencing using BID-seq uncovers abundant pseudouridines in mammalian mRNA at base resolution. Nat. Biotechnol. 41, 344–354 (2023).

(5) Xu, H. et al. Absolute quantitative and base-resolution sequencing reveals comprehensive landscape of pseudouridine across the human transcriptome. Nat. Methods 21, 2024–2033 (2024).

(6) Fang, X. et al. A bisulfite-assisted and ligation-based qPCR amplification technology for locus-specific pseudouridine detection at base resolution. Nucleic Acids Res. 52, (2024).

(7) Wyres, K. L., Lam, M. M. C. & Holt, K. E. Population genomics of Klebsiella pneumoniae. Nature Reviews Microbiology vol. 18 Preprint at https://doi.org/10.1038/s41579-019-0315-1 (2020).

(8) Kerr, K. G. & Snelling, A. M. *Pseudomonas aeruginosa*: a formidable and ever-present adversary. Journal of Hospital Infection vol. 73 Preprint at https://doi.org/10.1016/j.jhin.2009.04.020 (2009).

(9) Ehling-Schulz, M., Lereclus, D. & Koehler, T. M. The Bacillus cereus Group: Bacillus Species with Pathogenic Potential . Microbiol. Spectr. 7, (2019).

(10) Xin, X. F., Kvitko, B. & He, S. Y. Pseudomonas syringae: What it takes to be a pathogen. Nature Reviews Microbiology vol. 16 Preprint at https://doi.org/10.1038/nrmicro.2018.17 (2018).